# EuroBERT: Scaling Multilingual Encoders for European Languages

**Nicolas Boizard**[*1,3], **Hippolyte Gisserot-Boukhlef**[*2,3], **Duarte M. Alves**[*4,5],
**André Martins**[4,5,6], **Ayoub Hammal**[7,8,9], **Caio Corro**[8,10,11], **Céline Hudelot**[3],
**Emmanuel Malherbe**[2], **Etienne Malaboeuf**[12], **Fanny Jourdan**[13], **Gabriel Hautreux**[12],
**João Alves**[6], **Kevin El-Haddad**[1,17], **Manuel Faysse**[3,14], **Maxime Peyrard**[8,15],
**Nuno M. Guerreiro**[3,4,5,6], **Patrick Fernandes**[4,5,18], **Ricardo Rei**[6], **Pierre Colombo**[3,16]

[1]Diabolocom, [2]Artefact Research Center, [3]MICS, CentraleSupélec, Université
Paris-Saclay, [4]Instituto Superior Técnico & Universidade de Lisboa (Lisbon ELLIS Unit),
[5]Instituto de Telecomunicações, [6]Unbabel, [7]Université Paris-Saclay, [8]CNRS, [9]LISN,
[10]INSA Rennes, [11]IRISA, [12]CINES, [13]IRT Saint Exupéry, [14]Illuin Technology
[15]Université Grenoble Alpes, Grenoble INP, LIG, [16]Equall, [17]ISIA Lab,
[18]Carnegie Mellon University

{nicolas.boizard,hippolyte.gisserot-boukhlef}@centralesupelec.fr,
duartemalves@tecnico.ulisboa.pt

## Abstract

General-purpose multilingual vector representations, used in retrieval, regression, and classification, are traditionally obtained from bidirectional encoder models. Despite their wide applicability, encoders have been recently overshadowed by advances in generative decoder-only models. However, many innovations driving this progress are not inherently tied to decoders. In this paper, we revisit the development of multilingual encoders through the lens of these advances, and introduce EuroBERT, a family of multilingual encoders covering European and widely spoken global languages. Our models outperform existing alternatives across a diverse range of tasks, spanning multilingual capabilities, mathematics, and coding, and natively support sequences of up to 8,192 tokens. We also examine the design decisions behind EuroBERT, offering insights into our dataset composition and training pipeline. We publicly release the EuroBERT models,[1] including intermediate training checkpoints, together with our training framework.

## 1 Introduction

Many important tasks in Natural Language Processing (NLP), including information retrieval, classification, or regression, are built upon general-purpose vector representations. These representations are traditionally obtained from *bidirectional* encoder models, which aggregate information from the left and right contexts of each token (Devlin et al., 2019; Conneau et al., 2020; He et al., 2023). In contrast, recent advances in generative modeling have shifted the research community's attention towards *unidirectional* architectures (Bai et al., 2023; Llama Team, 2024; OLMo et al., 2025). Notably, these efforts have identified several key performance drivers that span architectural advances, data improvements, and increased scale. Yet, despite no apparent barrier to transferring these insights to *bidirectional* architectures, little effort has been devoted towards this objective, forcing practitioners to depend on outdated models.

---

*Equal contribution.

[1]https://huggingface.co/EuroBERT

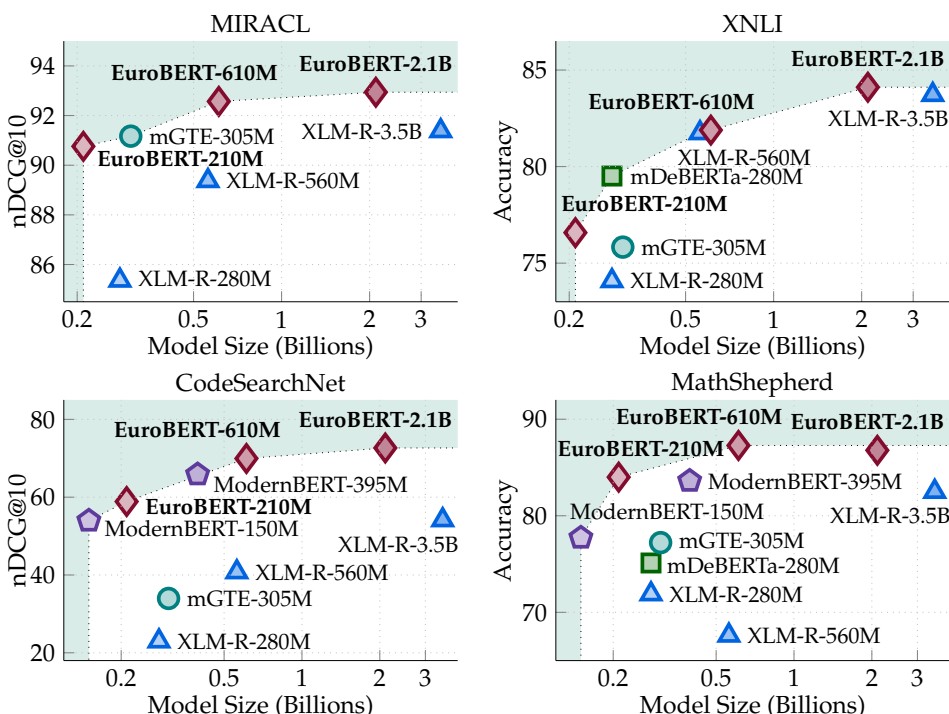

Figure 1: Pareto plots for multilingual tasks (top), showing retrieval performance on MIRACL and sentence classification on XNLI, and for math and code tasks (bottom), featuring CodeSearchNet and MathShepherd. The shaded regions indicate the Pareto frontiers.

In this paper, we introduce a refreshed recipe for training general-purpose multilingual encoders, resulting in the EuroBERT family. Drawing inspiration from recent progress in decoder models, our models feature an updated architecture (§2.1), and are trained on a 5T-token multilingual dataset, covering widely spoken European and global languages, along with mathematics and code (§2.2). We adopt a masked language modeling objective, and employ a two-phase training pipeline, adjusting the data distribution in the second training phase to improve downstream performance (§2.3).

We extensively evaluate the EuroBERT models, comparing with similarly sized alternatives across a suite of tasks representative of real-world encoder applications (§3). Our models match or exceed the performance of alternative models, such as XLM-RoBERTa (Conneau et al., 2020), mGTE-MLM (Zhang et al., 2024) and ModernBERT (Warner et al., 2024), on multilingual retrieval, classification and regression tasks, and outperform them on code and mathematics tasks (Figure 1).

In order to provide further insights into the methodologies involved in large-scale encoder training, we also examine the impact of our design choices through systematic ablations on several components of our annealing recipe (§4). We explore the choice of masking ratio, showing that while higher masking ratios benefit retrieval tasks, lower ratios improve sentence classification. Additionally, we highlight that including data for code and mathematics improves multilingual retrieval, but degrades classification accuracy. Contrary to expectations, we also observe that only selecting documents with high educational value degrades performance, which improves decoder LLMs (Penedo et al., 2024), and instead find that encoders benefit from a broader range of data sources.

Accompanying this work, we release the EuroBERT family, comprising three models with 210M, 610M and 2.1B parameters. To facilitate future research, we also release intermediate training checkpoints, as well as our training framework.

## 2 EuroBERT: A Refreshed Multilingual Encoder

The EuroBERT models incorporate design choices similar to the Llama 3 architecture (Llama Team, 2024) (§2.1). They are trained on a large multilingual corpus, which also includes code and mathematics (§2.2). Their training pipeline has two stages, pre-training and annealing, and employs the masked language modeling (MLM) objective (§2.3).

### 2.1 Architecture

The EuroBERT models are based on a standard dense transformer (Vaswani et al., 2017), with several architectural changes. Similarly to Llama 2 (Touvron et al., 2023), we remove all biases. Additionally, we incorporate grouped query attention (Ainslie et al., 2023), swish gated linear units (Shazeer, 2020), root mean square layer normalization (Zhang & Sennrich, 2019), and rotary position embeddings (Su et al., 2024). However, unlike decoder models, we do not apply causal masking.[2]

### 2.2 Dataset

To train EuroBERT, we construct a multilingual 5T-token corpus — 4.8T tokens for pre-training and 200B for annealing — which includes 15 languages: English, French, German, Spanish, Chinese, Italian, Russian, Polish, Portuguese, Japanese, Vietnamese, Dutch, Arabic, Turkish, and Hindi.[3] Following prior work on curriculum learning (Hu et al., 2024), we adjust the data distribution to emphasize higher-quality datasets during annealing.

**Pre-training mixture.** We use FineWeb (Penedo et al., 2024) for English, and CulturaX (Nguyen et al., 2024) for multilingual data. We also incorporate EuroLLM parallel data (Martins et al., 2024), which can improve cross-lingual transfer (Conneau & Lample, 2019; Reid & Artetxe, 2022; 2023), by concatenating to-English and from-English translation pairs, separated by a special `<|parallel_sep|>` token. Finally, inspired by the benefits of training on code for decoder models (Aryabumi et al., 2024), we add 38 programming languages from The Stack v2 (Lozhkov et al., 2024), and Proof-Pile-2 (Azerbayev et al., 2024) for mathematics, both of which we find improve multilingual retrieval (§4).

**Annealing mixture.** We classified data not seen during pre-training into four quality levels according to educational value using the multilingual classifier from Martins et al. (2024).[4] We then kept the documents above the third threshold which, contrary to our expectations, improved performance on downstream tasks. Additionally, we adjusted the data distribution based on multiple ablations. Specifically, we decreased English while proportionally increasing the remaining languages. We also decreased the amount of code and math while increasing parallel data (§4).[5]

### 2.3 Training Recipe

**Masked language modeling.** We pre-train EuroBERT models with a 50% masking ratio, following the insights from Wettig et al. (2023), who find that masking 15% and 30% of tokens is sub-optimal, and that larger models benefit from higher masking ratios. For the subsequent annealing, however, we lower the masking ratio to 10% based on downstream evaluations (§4), aligning with the findings from Yang et al. (2023) and Ankner et al. (2024).

**Hyperparameters.** We employed the Warmup-Stable-Decay (WSD) scheduler (Shen et al., 2024), with a linear warm-up phase of 2,000 steps, a constant learning rate of $1 \times 10^{-4}$

---

[2]We provide more architecture details in Appendix A.

[3]These languages were selected to balance European and widely spoken global languages, and ensure representation across diverse alphabets and language families.

[4]Similar to Penedo et al. (2024), the classifier groups documents into buckets based on their educational value, with higher numbers indicating higher quality.

[5]We provide further details on our pretraining and annealing datasets in Appendix C.

during pre-training, and a cosine scheduler decaying to 0 during the annealing phase. During pre-training, we packed sentences to 2,048 tokens and used a Rotary Position Embedding (RoPE) value of 10,000. In the annealing phase, we increased the RoPE theta to 250,000 and randomly cropped our training documents to lengths between 12 and 8,192 tokens. We adopted this approach because, due to pre-processing constraints, our training data had already been segmented into fixed-length documents, making standard variable-length training infeasible. Therefore, we introduced random cropping of these fixed-length sequences as an approximation of variable-length training. We found that this approach outperforms training only on fixed lengths (§4), further highlighting the necessity for variable length documents during long context training (Gao et al., 2024).

**Infrastructure.** We trained EuroBERT using 92 MI250X GPUs for EuroBERT-210M, 384 MI250X GPUs for EuroBERT-610M, and 96 MI300A GPUs for EuroBERT-2.1B, for a total of 200k GPU hours. Our training framework incorporates FlashAttention (Dao, 2023), fused cross-entropy from LigerKernel (Hsu et al., 2024), torch.compile (Ansel et al., 2024), and hybrid sharding with Fully Sharded Data Parallel (Zhao et al., 2023).

## 3 Evaluation

### 3.1 Evaluation Setup

**Datasets and tasks.** We select a suite of tasks to cover various real-world use cases for encoders. For multilingual tasks, we evaluate retrieval using MIRACL (Zhang et al., 2023), MLDR (Chen et al., 2024), WikipediaRetrieval[6], and CC-News (de Gibert et al., 2024). We assess sentence classification with XNLI (Conneau et al., 2018), PAWS-X (Yang et al., 2019), AmazonReviews (Keung et al., 2020) and MassiveIntent (Keung et al., 2020). Additionally, we evaluate token classification using the NER task from the XGLUE benchmark (Liang et al., 2020). We evaluate sequence regression on the WMT quality estimation task (Bojar et al., 2017; 2018; Barrault et al., 2019; 2020; Akhbardeh et al., 2021; Kocmi et al., 2022), and on summary evaluation using SeaHorse (Clark et al., 2023). For code-related tasks, we evaluate retrieval on CodeSearchNet (Husain et al., 2019) and DupStackMath (Hoogeveen et al., 2015), and classification on CodeDefect (Zhou et al., 2019) and CodeComplexity (Jeon et al., 2023). Finally, in the mathematical domain, we test retrieval on the MathFormula (Drechsel et al., 2025) task, and classification on MathShepherd (Wang et al., 2024b).[7]

**Baselines.** We compare with the multilingual XLM-RoBERTa (Conneau et al., 2020; Goyal et al., 2021), mGTE-MLM (Zhang et al., 2024)[8] and mDeBERTa-v3 (He et al., 2023). For code and mathematics, we also compare with the English-only ModernBERT (Warner et al., 2024).

**Fine-tuning.** For each task, models are trained for 10,000 steps (unless otherwise specified) on the corresponding training split using a batch size of 32, a 10% warm-up ratio, and a linear learning rate decay. For small datasets requiring multiple epochs, we apply early stopping with a patience of one epoch based on validation performance. To account for model specificities, we fine-tune using 10 logarithmically spaced learning rates ($1 \times 10^{-5}$ to $1 \times 10^{-4}$), selecting the one that achieves the highest validation metric.[9] For sequence classification, we use the cross-entropy loss during training, while for sequence regression, we substitute it with mean squared error.[10] For token classification tasks, we use the token-level cross-entropy loss, assigning each sub-token in an entity to the corresponding entity

---

[6] https://huggingface.co/datasets/Samoed/WikipediaRetrievalMultilingual

[7] We detail the evaluation setup for each task in Appendix D.

[8] Since the EuroBERT models are general-purpose encoders, we evaluate them against the pre-trained mGTE-MLM variant, which, similarly, was not optimized for retrieval tasks.

[9] For the baselines, we set additional fine-tuning hyperparameters according to the original paper. For EuroBERT models, we maintain the values from pre-training and annealing.

[10] On summarization (SeaHorse), we train for 5,000 steps to reduce computational costs.

| Benchmark | mDeBERTa 280M | mGTE 305M | XLM-RoBERTa 280M | XLM-RoBERTa 560M | XLM-RoBERTa 3.5B | EuroBERT 210M | EuroBERT 610M | EuroBERT 2.1B |
|---|---|---|---|---|---|---|---|---|
| **Retrieval** *(nDCG@10)* | | | | | | | | |
| MIRACL | 37.5 (6) | 91.2 (2) | 85.4 (4) | 89.4 (3) | 91.4 (2) | 90.8 (2) | 92.6 (2) | **92.9 (1)** |
| | *43.7 (5)* | *93.8 (2)* | *89.5 (4)* | *91.6 (3)* | *92.6 (3)* | *95.1 (1)* | *95.0 (1)* | *94.8 (1)* |
| MLDR | 18.3 (6) | 67.8 (2) | 54.6 (5) | 60.8 (4) | 65.9 (2) | 65.4 (3) | **68.6 (1)** | 66.1 (2) |
| | *20.0 (6)* | *73.2 (2)* | *58.7 (5)* | *65.2 (4)* | *70.0 (3)* | *73.4 (2)* | *75.8 (1)* | *72.9 (2)* |
| CC-News | 18.5 (7) | 71.3 (4) | 61.6 (6) | 72.8 (3) | **80.9 (1)** | 67.2 (5) | 75.6 (2) | 75.9 (2) |
| | *15.8 (7)* | *71.5 (4)* | *60.4 (6)* | *72.1 (4)* | *80.9 (1)* | *69.0 (4)* | *76.6 (2)* | *76.9 (2)* |
| Wikipedia | 57.6 (5) | 94.1 (2) | 91.0 (4) | 93.1 (3) | **96.3 (1)** | 94.4 (2) | **95.9 (1)** | 95.8 (1) |
| | *58.9 (5)* | *94.6 (3)* | *91.7 (4)* | *93.6 (3)* | *96.7 (1)* | *95.6 (2)* | *96.6 (1)* | *96.6 (1)* |
| **Sequence Classification** *(Accuracy)* | | | | | | | | |
| XNLI | 79.5 (4) | 75.8 (5) | 74.1 (6) | 81.7 (2) | **83.7 (1)** | 76.6 (5) | 81.9 (3) | **84.1 (1)** |
| | *82.0 (4)* | *78.4 (6)* | *76.6 (7)* | *84.1 (3)* | *86.1 (2)* | *79.9 (5)* | *84.7 (2)* | *86.8 (1)* |
| PAWS-X | 91.9 (2) | 89.8 (3) | 88.9 (4) | 92.4 (2) | **92.9 (1)** | 89.9 (3) | 92.2 (2) | **93.0 (1)** |
| | *91.9 (2)* | *89.8 (3)* | *88.9 (4)* | *92.4 (2)* | *92.9 (1)* | *89.9 (3)* | *92.2 (2)* | *93.0 (1)* |
| AmazonReviews | 62.1 (2) | 61.5 (3) | 61.1 (3) | **63.1 (1)** | **63.6 (1)** | 61.7 (2) | 62.6 (2) | **63.2 (1)** |
| | *63.7 (2)* | *62.7 (3)* | *62.7 (3)* | *64.5 (1)* | *64.7 (1)* | *63.0 (2)* | *64.0 (2)* | *64.5 (1)* |
| MassiveIntent | 86.5 (3) | 86.9 (2) | 86.3 (3) | **88.2 (1)** | 87.9 (1) | 86.5 (3) | 87.2 (2) | 87.5 (2) |
| | *87.3 (2)* | *87.5 (2)* | *87.2 (2)* | *88.8 (1)* | *88.5 (1)* | *87.2 (2)* | *87.8 (2)* | *88.2 (2)* |
| **Token Classification** *(F1 Score)* | | | | | | | | |
| NER | **96.2 (2)** | 95.2 (6) | 95.5 (5) | **96.1 (2)** | 96.3 (2) | 94.7 (6) | 95.9 (4) | 95.2 (6) |
| | *96.2 (2)* | *95.2 (6)* | *95.5 (5)* | *96.1 (2)* | *96.3 (2)* | *94.7 (6)* | *95.9 (4)* | *95.2 (6)* |
| **Sequence Regression** *(Spearman)* | | | | | | | | |
| WMT (Ref-based) | 45.7 (4) | 43.9 (5) | 43.0 (6) | 45.3 (4) | **47.7 (2)** | 45.2 (4) | 46.0 (3) | **47.3 (2)** |
| | *46.5 (4)* | *44.0 (5)* | *43.1 (6)* | *45.6 (5)* | *48.5 (2)* | *45.1 (4)* | *46.5 (4)* | *48.5 (2)* |
| WMT (Ref-free) | 42.0 (3) | 38.5 (5) | 36.5 (7) | 40.8 (4) | **44.5 (1)** | 41.0 (3) | 41.5 (3) | 38.7 (5) |
| | *41.6 (2)* | *37.7 (5)* | *34.2 (6)* | *39.0 (4)* | *44.4 (1)* | *40.5 (3)* | *41.1 (3)* | *38.8 (4)* |
| SeaHorse | 64.2 (5) | 63.0 (6) | 61.1 (7) | 65.5 (4) | 67.5 (2) | 63.8 (5) | 66.0 (3) | **67.5 (1)** |
| | *60.3 (5)* | *59.2 (6)* | *56.9 (7)* | *61.4 (4)* | *63.3 (2)* | *60.1 (5)* | *62.7 (2)* | *64.0 (1)* |

Table 1: Results for multilingual tasks, with scores aggregating all languages shown above and scores aggregating European languages in *italic* below. Models are grouped into statistically significant clusters, with best ranked models highlighted in bold.

label.[11] For all retrieval tasks, we finetune for 1,000 steps on MS-MARCO (Bajaj et al., 2016)[12] using the InfoNCE loss (Oord et al., 2018) with in-batch negatives and cosine similarity.

**Evaluation metrics.** We report accuracy for sequence classification, Spearman rank correlation for regression, F1 score for token classification, and nDCG@10 for retrieval tasks. We also follow Freitag et al. (2023), and group systems into language-specific clusters based on statistically significant performance gaps at 95% confidence thresholds. We then compute system-level rankings using a normalized Borda count (Colombo et al., 2022), defined as the average over the obtained per-language clusters. Note that a first cluster will only exist if a model significantly outperforms all others on a majority of languages.

### 3.2 Results

Table 1 presents the aggregated results across multilingual tasks, and Table 2 summarizes performance on code and mathematics benchmarks.[13]

---

[11]At inference time, the final label of an entity is determined by majority vote over the sub-tokens.

[12]Since many retrieval datasets lack dedicated training splits, we use MS-MARCO, an English-only dataset. This choice also allows us to assess cross-lingual generalization.

[13]We provide per-language results in Appendix F.

| Benchmark | ModernBERT | | mDeBERTa | mGTE | XLM-RoBERTa | | | EuroBERT | | |
|---|---|---|---|---|---|---|---|---|---|---|
| | 150M | 395M | 280M | 305M | 280M | 560M | 3.5B | 210M | 610M | 2.1B |
| **Code Retrieval** *(nDCG@10)* | | | | | | | | | | |
| CodeSearchNet | 53.9 5 | 65.8 3 | 2.8 | 34.0 7 | 23.0 8 | 40.8 6 | 54.1 5 | 58.9 4 | 69.9 **2** | **72.6 1** |
| DupStackMath | 39.7 4 | 45.5 2 | 10.2 7 | 37.5 4 | 29.3 6 | 36.9 5 | 42.9 3 | 41.7 3 | 46.0 2 | **48.3 1** |
| **Code Classification** *(Accuracy)* | | | | | | | | | | |
| CodeComplexity | 86.1 3 | 88.6 3 | 73.9 5 | 74.5 5 | 74.1 5 | 83.6 4 | 84.3 4 | 91.9 2 | **94.2 1** | **95.2 1** |
| CodeDefect | 65.8 3 | 67.0 2 | 64.7 3 | 63.5 4 | 61.9 4 | 54.3 5 | 65.8 3 | **69.5 1** | **69.0 1** | 67.7 2 |
| **Math Retrieval** *(nDCG@10)* | | | | | | | | | | |
| MathFormula | 89.6 5 | 91.9 2 | 85.2 7 | 83.4 8 | 83.1 8 | 81.4 | 89.1 6 | 91.5 3 | **92.6 1** | 91.0 4 |
| **Math Classification** *(Accuracy)* | | | | | | | | | | |
| MathShepherd | 77.7 4 | 83.6 2 | 75.1 5 | 77.2 4 | 71.9 6 | 67.6 7 | 82.5 3 | 84.0 2 | **87.3 1** | **86.8 1** |

Table 2: Results for code and mathematical tasks. Models are grouped into statistically significant clusters, with best ranked models highlighted in bold.

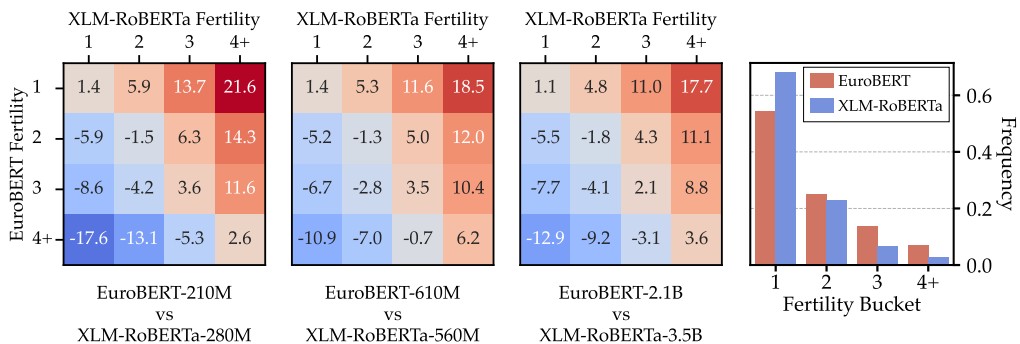

Figure 2: Difference in F1 Score between EuroBERT and XLM-RoBERTa by tokenizer fertility (left plot) and fertility distribution for both tokenizers (right plot) on the NER dataset.

**The EuroBERT family delivers strong performance across diverse domains and tasks.** Our largest model, EuroBERT-2.1B, ranks first on 10 out of 18 tasks, competing closely with the larger XLM-RoBERTa-3.5B. EuroBERT-610M is also on par with XLM-RoBERTa-3.5B across several multilingual tasks while being five times smaller, and outperforms it on code and mathematics benchmarks. Likewise, EuroBERT-210M matches XLM-RoBERTa-560M performance while having less than half the parameters, and consistently outperforms other models of similar size, showing especially strong results on European languages.

**EuroBERT is effective at document ranking.** Across domains, EuroBERT consistently ranks high for retrieval tasks. Notably, the 210M and 610M models outperform all alternatives of comparable sizes, and are competitive with the larger XLM-RoBERTa-3.5B.[14]

**EuroBERT models are on par with similarly sized models for sequence classification.** On sequence classification, no model significantly outperforms all others. During the development of EuroBERT, we found that several design decisions lead to a trade-off between retrieval and classification capabilities (§4). We highlight, however, that EuroBERT-2.1B is still among the highest ranking systems, and that the smaller models in the family are competitive with models of comparable size.

---

[14]For retrieval, increasing model size did not always lead to better results. Further analysis, in Appendix E, revealed that EuroBERT-2.1B benefits significantly from a more thorough grid search.

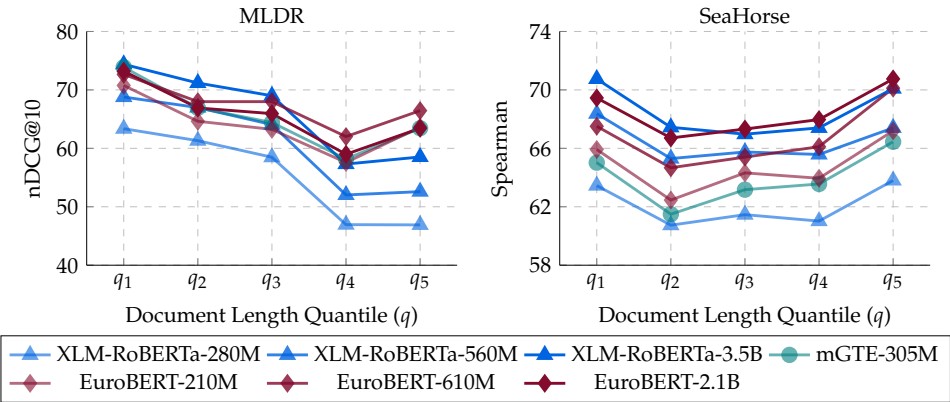

Figure 3: Results by length of the positive documents for retrieval (MLDR) and input documents for summarization (SeaHorse).

**There is room for improvement on token classification.** On the NER task, EuroBERT lags behind XLM-RoBERTa. However, we observed that models achieve comparable F1 scores when splitting entities into a similar number of tokens, as shown in Figure 2. In contrast, models perform significantly worse when segmenting entities into a larger number of tokens. We posit that token classification tasks may benefit from larger vocabularies, with lower fertility, such as the one used in the XLM-RoBERTa family. We also note, however, that increasing the vocabulary size also increases the number of parameters, particularly for smaller models.[15]

**EuroBERT can function as an evaluation metric.** EuroBERT models match or exceed the performance of similarly sized systems in reference-based translation evaluation. For reference-free evaluation, while EuroBERT-2.1B lags behind the larger XLM-RoBERTa, the 210M and 610M variants are competitive with other baselines. In the future, we will explore other training signals to further enhance EuroBERT's cross-lingual capabilities. For summary evaluation, EuroBERT models consistently outperform similarly sized alternatives.

**EuroBERT maintains performance at longer context lengths.** Figure 3 compares the long context performance of EuroBERT and XLM-RoBERTa. On both retrieval and summary evaluation, EuroBERT maintains performance at longer contexts, whereas XLM-RoBERTa suffers notable degradation.

**The EuroBERT family performs strongly in tasks related to code and mathematics.** On these tasks in the code and math domain, all EuroBERT models consistently surpass other systems. Notably, EuroBERT-210M reflects most of the performance of the larger models in the family, and ranks above all baselines, highlighting its capabilities at a smaller scale.

## 4  Training Recipe Analysis

We measure the impact of various design decisions made during the development of EuroBERT with extensive ablations. Following Blakeney et al. (2024) and Llama Team (2024), we perform multiple annealing runs on 40B tokens, each varying a different component of our recipe, and measure the performance on the XNLI and MIRACL validation sets, the former representing multilingual classification and the latter multilingual retrieval.[16]

---

[15]For example, doubling the vocabulary size of EuroBERT-210M would add 100M parameters to the model embeddings.

[16]We follow the procedure from §3, but instead evaluate on the validation splits considering only European languages.

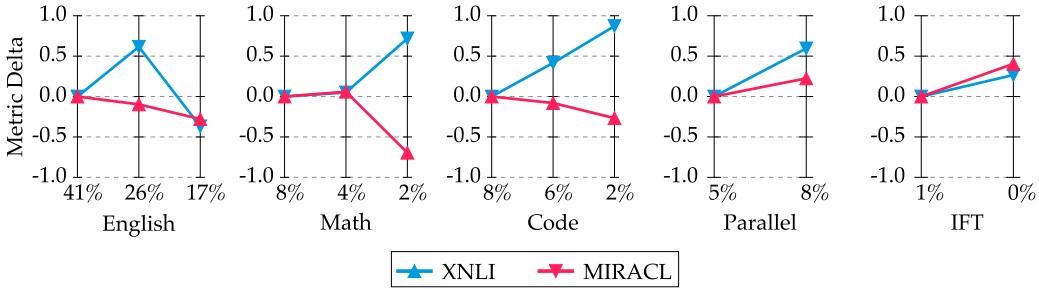

Figure 4: Impact of changing data subset ratios during annealing. The first vertical axis in each subplot denotes the *reference* data mix from Table 6.

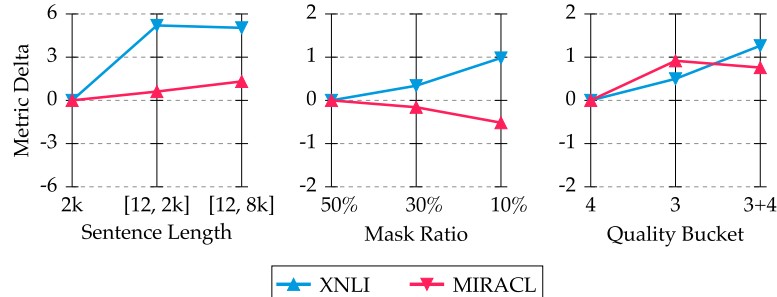

Figure 5: Impact of hyperparameter choices during annealing. The first vertical axis in each subplot denotes the *reference* data mix from Table 6.

**Balancing the language distribution enhances performance.** The left-most plot in Figure 4 reports retrieval and classification performance as the proportion of English is reduced and re-distributed between other languages. We observe that a more balanced distribution improves overall results. However, when the language distribution becomes too close to uniform, there is a degradation in performance.

**Including math and code improves multilingual retrieval, but degrades multilingual classification.** The second and third plots in Figure 4 show MIRACL performance dropping and XNLI accuracy rising as the quantity of math and code data decreases. In future work, we will investigate how to better balance downstream task performance during pre-training.

**Increasing parallel data improves multilingual classification and retrieval.** The forth plot in Figure 4 presents XNLI and MIRACL performance when increasing parallel data. In line with recent work showing the benefits of pre-training with parallel data (Anil et al., 2023; Briakou et al., 2023; Alves et al., 2024), we find it improves both benchmarks.

**Adding instruction fine-tuning data degrades model performance.** The right-most plot in Figure 4 analyses the impact of adding instructions during annealing, which can improve performance for decoder language models (Wei et al., 2022; Chung et al., 2024). In contrast to decoders, it leads to worse performance when training an encoder model.

**Varying sentence length improves performance.** The first plot in Figure 5 examines the impact of variable sentence lengths during annealing. Compared to the fixed packed sentence lengths employed in pretraining, variable sentence lengths significantly boosts XNLI and moderately MIRACL performance.[17] This improvement remains stable, without degradation when the maximum context length is extended to 8,192 tokens.

---

[17]We hypothesize this gap stems from the prevalence of shorter sequences in the XNLI dataset.

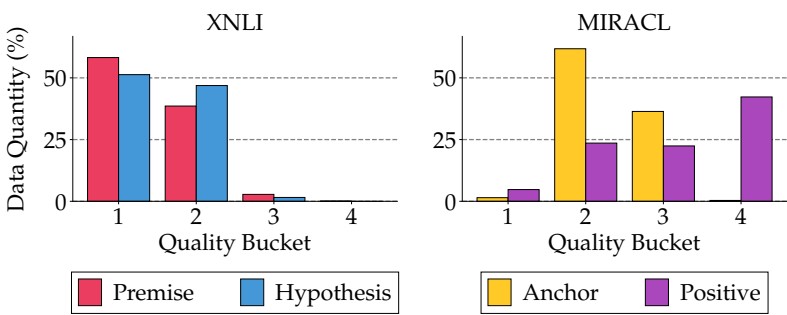

Figure 6: Quality buckets for XNLI and MIRACL English training subset.

**A reduced masking ratio during annealing enhances classification accuracy.** Similarly to previous research advocating a lower masking ratio in later training (Yang et al., 2023; Ankner et al., 2024), we also find that reducing it to 10% during the annealing phase improves EuroBERT's accuracy on XNLI, though it leads to a decline in MIRACL scores.

**Filtering data based on educational value can degrade results.** Contrary to initial expectations, using the highest quality data bucket during annealing does not result in better performance on XNLI and MIRACL. Instead, as illustrated in the right-most plot of Figure 5, mixing the buckets with quality levels 3 and 4 leads to the best overall results. Curiously, inspecting the evaluation splits of these datasets revealed that our quality filter would discard nearly all examples in XNLI, and many in MIRACL, as shown in Figure 6. This result highlights a potential domain mismatch, wherein our training data deviates from the distribution of downstream tasks. Indeed, while educational value fits assistant-like tasks typically delegated to LLMs, a broader coverage of textual data, reflected in mixing both quality buckets, may be more inline with general-purpose vector representations. In future work, we would like to explore quality filters that are better tailored to encoders.

**Final annealing configuration.** The previous results revealed several design choices that trade off classification and retrieval performance. In the final data mix, we aimed to balance these two tasks. Based on the previous analysis, we created our final annealing dataset by selecting data above the third threshold. We reduced the proportion of English to 26% while proportionally increasing the share of the remaining languages. We allocated 6% and 4% of the data mix to math and code, respectively. Additionally, we increased the proportion of parallel data to 6%, and removed instructions. We finally lowered the masking ratio to 10% and performed annealing with random sentence lengths of up to 8,192 tokens.

## 5 Related Work

Encoder models have shown strong performance in non-generative tasks, such as classification and retrieval (Devlin et al., 2019; Liu et al., 2019; He et al., 2023; Acheampong et al., 2021; Ma et al., 2019; Karpukhin et al., 2020; Wang et al., 2024a). Variants of these models have also extended support to multiple languages and cross-lingual tasks (Conneau et al., 2020). However, scaling to many languages introduces the "curse of multilinguality" (Conneau et al., 2020; Chang et al., 2024), where interference across languages degrades performance. Notably, increasing model capacity has been shown to mitigate this effect (Conneau et al., 2020), motivating our focus on scale.

Encoders are typically trained with masked language modeling (MLM) (Devlin et al., 2019). While alternatives like replaced token detection (He et al., 2023) exist, we adopt the MLM objective because initial evaluations of existing models showed more balanced results across tasks. Building upon the effectiveness of higher masking ratios (Wettig et al., 2023), we mask 50% of the training tokens during pre-training. Prior work has also shown the benefits of decreasing the masking ratio in later phases of training (Yang et al., 2023; Ankner et al., 2024),

we also lower our masking ratio to 10% during annealing. Interestingly, we demonstrate that this choice improves classification accuracy, but reduces retrieval quality.

Recent concurrent work, such as ModernBERT (Warner et al., 2024) and mGTE (Zhang et al., 2024), also revisits encoders in light of advances in decoder models. Similar to our approach, they incorporate grouped query attention (Ainslie et al., 2023), rotary positional embeddings (Su et al., 2024), gated linear units (Shazeer, 2020), root mean square layer normalization (Zhang & Sennrich, 2019), and support for longer context windows. However, we additionally draw inspiration from Llama Team (2024); Yang et al. (2024) by including code and mathematical data during pre-training, which we show improves retrieval quality.

## 6   Conclusion

We propose a recipe for training general-purpose multilingual encoders, creating the EuroBERT family. We incorporate recent architectural advances from decoder models, and train on a multilingual dataset containing European and globally spoken languages, together with code and mathematics. Our models outperform existing alternatives on a comprehensive suite of tasks covering multilingual capabilities, mathematics and code. We also extensively analyze the design decisions behind EuroBERT's dataset and training pipeline. Alongside this paper, we release all models in the EuroBERT family, including intermediate training checkpoints, and our training framework to facilitate future research.

## Acknowledgments

We sincerely thank the ADASTRA supercomputer (CINES) for its technical support and high-performance computing (HPC) resources, provided through grants C1615122 and GDA2401. We also appreciate the support of the French government through the France 2030 program as part of the ArGiMi project. This work was also supported by the EU's Horizon Europe Research and Innovation Actions (UTTER, contract 101070631), by the project DECOLLAGE (ERC-2022-CoG 101088763), by the Portuguese Recovery and Resilience Plan through project C645008882-00000055 (Center for Responsible AI), and by FCT/MECI through national funds and when applicable co-funded EU funds under UID/50008: Instituto de Telecomunicações. Duarte was also partially supported by the DataIA Institute, whose contributions facilitated the completion of this work.

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

# A  EuroBERT Model Architecture

Table 3 reports the architectural details of the EuroBERT model family.

| Model Size | 210M | 610M | 2.1B |
|---|---|---|---|
| Layers | 12 | 26 | 32 |
| Embedding Dimension | 768 | 1,152 | 2,304 |
| FFN Dimension | 3,072 | 4,096 | 6,144 |
| Attention Heads | 12 | 18 | 18 |
| Key/Value Heads | 12 | 6 | 6 |
| Layer Normalization | RMSNorm | | |
| RMSNorm $\epsilon$ | $1 \times 10^{-5}$ | | |
| Activation Function | SwiGLU | | |
| Vocabulary Size | 128,000 | | |
| Positional Embeddings | RoPE | | |
| RoPE $\theta$ | 250,000 | | |
| Tokenizer | LLaMA 3 | | |

Table 3: Summary of architectural hyperparameters for EuroBERT models of different sizes.

# B  Training Details

We trained the EuroBERT family utilizing 92 MI250X GPUs for EuroBERT-210M over 15k hours, 384 MI250X GPUs for EuroBERT-610M over 92k hours, and 96 MI300A GPUs for EuroBERT-2.1B over 106k hours, hyperparameter selections are detailed in Table 4. We find this training recipe highly stable, with no loss spikes or need for intervention to address model training divergence (Figure 7).

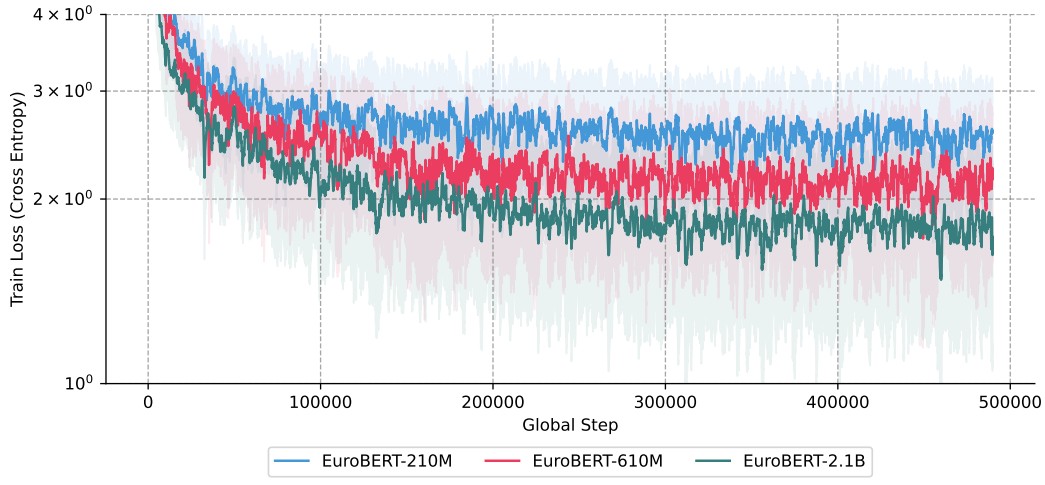

Figure 7: Pre-training Loss for all EuroBERT models on a logarithmic scale.

| Parameter | 210M | 610M | 2.1B |
|---|---|---|---|
| **Pre-training** | | | |
| LR | | 1e-4 | |
| LR Scheduler | | WSD | |
| Warmup Steps | | 2,000 | |
| Context Length | | 2,048 | |
| Weight Initialisation | | $\mathcal{N}(\mu = 0, \sigma^2 = 0.2)$ | |
| **Annealing** | | | |
| LR | | 1e-4 to 0 | |
| LR Scheduler | | Cosine | |
| Context Length | | 8,192 | |
| **Optimizer** | | | |
| Optimizer | | AdamW | |
| Beta1 | | 0.9 | |
| Beta2 | | 0.95 | |
| Epsilon (eps) | | 1e-5 | |
| Weight Decay | | 0.1 | |
| Clip Grad Norm | | 1.0 | |
| **Training Setup** | | | |
| Per-GPU Batch Size | 24 | 12 | 10 |
| Gradient Accumulation Steps | 1 | 1 | 5 |
| GPUs | 192 | 384 | 96 |
| Tokens/Step | 9,437,184 | 9,437,184 | 9,830,400 |

Table 4: Training hyperparameters for EuroBERT models (210M, 610M, 2.1B). The optimizer and Tokens/Step remain consistent across both pre-training and annealing phases.

## C   Data Mix

Table 5 details the data sources used throughout training, as well as the number of tokens used from each of the data source during pre-training. Table 6 specifies the different annealing mixes used for ablations.

| Source | Subset | Tokens (M) | Mix (%) | Source | Subset | Tokens (M) | Mix (%) |
|---|---|---|---|---|---|---|---|
| FineWeb | English | 2,002,327 | 41.34 | The-Stack v2 | JavaScript | 58,440 | 1.21 |
| CulturaX | French | 295,113 | 6.09 | The-Stack v2 | PHP | 25,620 | 0.53 |
| CulturaX | German | 291,514 | 6.02 | The-Stack v2 | C# | 24,842 | 0.51 |
| CulturaX | Spanish | 290,489 | 6.00 | The-Stack v2 | Python | 21,521 | 0.44 |
| CulturaX | Chinese | 238,467 | 4.92 | The-Stack v2 | Java | 20,950 | 0.43 |
| CulturaX | Italian | 120,128 | 2.48 | The-Stack v2 | Go | 14,766 | 0.30 |
| CulturaX | Russian | 116,797 | 2.41 | The-Stack v2 | TypeScript | 11,307 | 0.23 |
| CulturaX | Portuguese | 112,321 | 2.32 | The-Stack v2 | HTML | 7,962 | 0.16 |
| CulturaX | Japanese | 112,242 | 2.32 | The-Stack v2 | Lua | 7,733 | 0.16 |
| CulturaX | Polish | 111,659 | 2.31 | The-Stack v2 | Ruby | 5,524 | 0.11 |
| CulturaX | Turkish | 53,126 | 1.10 | The-Stack v2 | Vue | 5,411 | 0.11 |
| CulturaX | Arabic | 52,413 | 1.08 | The-Stack v2 | R | 5,287 | 0.11 |
| CulturaX | Vietnamese | 50,661 | 1.05 | The-Stack v2 | Shell | 4,793 | 0.10 |
| CulturaX | Dutch | 50,646 | 1.05 | The-Stack v2 | Swift | 3,766 | 0.08 |
| CulturaX | Hindi | 25,544 | 0.53 | The-Stack v2 | reStructuredText | 3,761 | 0.08 |
| EuroLLM Parallel | es ↔ en | 50,613 | 1.05 | The-Stack v2 | JSON | 3,586 | 0.07 |
| EuroLLM Parallel | fr ↔ en | 44,891 | 0.93 | The-Stack v2 | Rust | 3,152 | 0.07 |
| EuroLLM Parallel | de ↔ en | 30,541 | 0.63 | The-Stack v2 | YAML | 2,716 | 0.06 |
| EuroLLM Parallel | it ↔ en | 18,702 | 0.39 | The-Stack v2 | Dart | 2,678 | 0.06 |
| EuroLLM Parallel | ru ↔ en | 13,808 | 0.29 | The-Stack v2 | RMarkdown | 2,058 | 0.04 |
| EuroLLM Parallel | nl ↔ en | 12,666 | 0.26 | The-Stack v2 | HCL | 1,423 | 0.03 |
| EuroLLM Parallel | pl ↔ en | 7,280 | 0.15 | The-Stack v2 | PowerShell | 1,027 | 0.02 |
| EuroLLM Parallel | ar ↔ en | 6,414 | 0.13 | The-Stack v2 | VBA | 1,027 | 0.02 |
| EuroLLM Parallel | zh ↔ en | 6,206 | 0.13 | The-Stack v2 | AsciiDoc | 970 | 0.02 |
| EuroLLM Parallel | cs ↔ en | 5,458 | 0.11 | The-Stack v2 | Groovy | 540 | 0.01 |
| EuroLLM Parallel | hu ↔ en | 4,599 | 0.09 | The-Stack v2 | CUDA | 406 | 0.01 |
| EuroLLM Parallel | vi ↔ en | 3,395 | 0.07 | The-Stack v2 | Dockerfile | 281 | 0.01 |
| EuroLLM Parallel | tr ↔ en | 2,975 | 0.06 | The-Stack v2 | Cython | 103 | 0.01 |
| EuroLLM Parallel | ja ↔ en | 2,687 | 0.06 | The-Stack v2 | COBOL | 96 | 0.01 |
| EuroLLM Parallel | hi ↔ en | 1,136 | 0.02 | The-Stack v2 | GraphQL | 83 | 0.01 |
| Proof-pile-2 | Arxiv | 121,503 | 2.51 | The-Stack v2 | HTTP | 82 | 0.01 |
| Proof-pile-2 | Open-Web-Math | 54,168 | 1.12 | The-Stack v2 | ABAP | 71 | 0.01 |
| Proof-pile-2 | Algebraic-stack | 35,985 | 0.74 | The-Stack v2 | RDoc | 16 | 0.01 |
| The-Stack v2 | C++ | 120,085 | 2.48 | The-Stack v2 | Metal | 8 | 0.01 |
| The-Stack v2 | SQL | 75,348 | 1.56 | The-Stack v2 | AppleScript | 7 | 0.01 |
| The-Stack v2 | C | 59,404 | 1.23 | **Total** | | **4,843,357** | **100** |

Table 5: Pre-training data, with a total of 4.8 trillion tokens (as measured by EuroBERT's tokenizer). We report the list of all dataset names and subsets used, including the number of tokens selected and their proportion in the final data mix.

| Mix | en | fr | de | nl | hi | it | ja | pl | pt | ru | es | ar | zh | tr | Code | Math | Parallel | IFT |
|---|---|---|---|---|---|---|---|---|---|---|---|---|---|---|---|---|---|---|
| Reference | 46.3 | 5.8 | 5.7 | 1.0 | 0.3 | 1.5 | 0.8 | 1.0 | 1.4 | 1.0 | 5.7 | 0.4 | 4.7 | 1.0 | 8.7 | 8.2 | 5.2 | 1.2 |
| English 26% | 26.0 | 6.0 | 6.0 | 4.0 | 4.0 | 4.0 | 4.0 | 4.0 | 4.0 | 4.0 | 6.0 | 4.0 | 6.0 | 4.0 | 4.0 | 4.0 | 5.0 | 1.0 |
| English 17% | 17.0 | 6.0 | 6.0 | 5.0 | 5.0 | 5.0 | 5.0 | 5.0 | 5.0 | 5.0 | 6.0 | 5.0 | 6.0 | 5.0 | 4.0 | 4.0 | 5.0 | 1.0 |
| Math 4% | 46.3 | 5.8 | 5.7 | 1.0 | 0.3 | 1.5 | 0.8 | 1.0 | 1.4 | 1.0 | 5.7 | 0.4 | 4.7 | 1.0 | 8.7 | 4.0 | 5.2 | 1.2 |
| Math 2% | 46.3 | 5.8 | 5.7 | 1.0 | 0.3 | 1.5 | 0.8 | 1.0 | 1.4 | 1.0 | 5.7 | 0.4 | 4.7 | 1.0 | 8.7 | 2.0 | 5.2 | 1.2 |
| Code 8% | 46.3 | 5.8 | 5.7 | 1.0 | 0.3 | 1.5 | 0.8 | 1.0 | 1.4 | 1.0 | 5.7 | 0.4 | 4.7 | 1.0 | 6.0 | 8.2 | 5.2 | 1.2 |
| Code 4% | 46.3 | 5.8 | 5.7 | 1.0 | 0.3 | 1.5 | 0.8 | 1.0 | 1.4 | 1.0 | 5.7 | 0.4 | 4.7 | 1.0 | 4.0 | 8.2 | 5.2 | 1.2 |
| Code 2% | 46.3 | 5.8 | 5.7 | 1.0 | 0.3 | 1.5 | 0.8 | 1.0 | 1.4 | 1.0 | 5.7 | 0.4 | 4.7 | 1.0 | 2.0 | 8.2 | 5.2 | 1.2 |
| Parallel 8% | 46.3 | 5.8 | 5.7 | 1.0 | 0.3 | 1.5 | 0.8 | 1.0 | 1.4 | 1.0 | 5.7 | 0.4 | 4.7 | 1.0 | 8.7 | 8.2 | 8.0 | 1.2 |
| IFT 0% | 46.3 | 5.8 | 5.7 | 1.0 | 0.3 | 1.5 | 0.8 | 1.0 | 1.4 | 1.0 | 5.7 | 0.4 | 4.7 | 1.0 | 8.7 | 8.2 | 5.2 | 0.0 |

Table 6: Data mix employed in the ablation study measuring the importance of different data subsets in the EuroBERT annealing phase.

# D   Details of Evaluation Datasets

This appendix offers additional details on the datasets used for evaluation. Table 7 presents the language coverage of all evaluation datasets, and below are additional specifications on the evaluated tasks.

| Task | European Languages | | | | | | | | Extra-European Languages | | | | | | | Code | Math |
|---|---|---|---|---|---|---|---|---|---|---|---|---|---|---|---|---|---|
| | en | de | es | fr | it | nl | pl | pt | ar | hi | ja | ru | tr | vi | zh | | |
| *Information Retrieval* | | | | | | | | | | | | | | | | | |
| MIRACL | ✓ | | ✓ | ✓ | | | | | | ✓ | ✓ | ✓ | ✓ | | | ✓ | |
| MLDR | ✓ | ✓ | ✓ | ✓ | ✓ | | | ✓ | | ✓ | ✓ | ✓ | | | | ✓ | |
| Wikipedia | ✓ | ✓ | | | ✓ | ✓ | | ✓ | | | | ✓ | | | | | |
| CC-News | ✓ | ✓ | ✓ | ✓ | ✓ | ✓ | ✓ | ✓ | ✓ | ✓ | ✓ | ✓ | ✓ | ✓ | ✓ | | |
| CodeSearchNet | ✓ | | | | | | | | | | | | | | | ✓ | |
| DupStackMath | ✓ | | | | | | | | | | | | | | | ✓ | |
| MathFormula | ✓ | | | | | | | | | | | | | | | | ✓ |
| *Sequence Classification* | | | | | | | | | | | | | | | | | |
| XNLI | ✓ | ✓ | ✓ | ✓ | | | | | ✓ | ✓ | | ✓ | ✓ | ✓ | ✓ | | |
| PAWS-X | ✓ | ✓ | ✓ | ✓ | | | | | | | ✓ | | | | ✓ | | |
| AmazonReviews | ✓ | ✓ | ✓ | ✓ | | | | | | | ✓ | | | | ✓ | | |
| MassiveIntent | ✓ | ✓ | ✓ | ✓ | ✓ | ✓ | ✓ | ✓ | ✓ | ✓ | ✓ | ✓ | ✓ | ✓ | ✓ | | |
| CodeDefect | ✓ | | | | | | | | | | | | | | | ✓ | |
| CodeComplexity | ✓ | | | | | | | | | | | | | | | ✓ | |
| MathShepherd | ✓ | | | | | | | | | | | | | | | | ✓ |
| *Sequence Regression* | | | | | | | | | | | | | | | | | |
| WMT | ✓ | ✓ | | ✓ | | ✓ | | | | ✓ | ✓ | ✓ | | | ✓ | | |
| SeaHorse | ✓ | ✓ | ✓ | | | | | | | | | | ✓ | ✓ | | | |
| *Token Classification* | | | | | | | | | | | | | | | | | |
| NER | ✓ | ✓ | ✓ | | | ✓ | | | | | | | | | | | |

Table 7: Language coverage across evaluation datasets.

**Retrieval datasets:**

- **MS-MARCO** (Bajaj et al., 2016) — English-only retrieval dataset used for fine-tuning, where each anchor-positive pair includes a mined hard negative, forming a triplet structure.

- **MIRACL** (Zhang et al., 2023) — Multilingual retrieval dataset. We use the semi-supervised version with labeled positive pairs provided by SentenceTransformers[18] as the primary data source. Anchors serve as queries, and the corpus consists of all positive documents in the dataset. Since only a single data split is available, we create validation and test sets by partitioning 50% of the original split for each, using queries as the split key to ensure no data leakage.

- **MLDR** (Chen et al., 2024) — Long-context multilingual retrieval dataset. As with MIRACL, we use the triplet version provided by SentenceTransformers and apply the same validation-test split strategy.

- **Wikipedia**[19] — Multilingual information retrieval dataset. Since only a single data split is available, we partition 50% of the queries into validation and test sets.

- **CC-News** (de Gibert et al., 2024) — Highly multilingual retrieval dataset. As with MIRACL, we use the SentenceTransformers dataset version as the primary data source and apply the same test-validation split method.

- **CodeSearchNet** (Husain et al., 2019) — Code retrieval dataset with comment-code query-positive pairs (SentenceTransformers version), processed similarly to the previous datasets.

---

[18] https://huggingface.co/collections/sentence-transformers/embedding-model-datasets-6644d7a3673a511914aa7552

[19] https://huggingface.co/datasets/Samoed/WikipediaRetrievalMultilingual

- **DupStackMath** (Hoogeveen et al., 2015) — Code retrieval dataset with queries, a corpus, and relevant documents, processed the same way as the above datasets.

- **MathFormula** (Drechsel et al., 2025) — Mathematical retrieval dataset consisting of pairs of equivalent formulas. The original dataset contains formula pairs labeled as true or false based on their equivalence, spanning 71 well-known mathematical formulas. To construct the retrieval dataset, we extract only equivalent formula pairs, retaining positive instances. Due to the dataset's large size, we sample 100 positive pairs per formula type for both validation and test sets. The final dataset is processed following the same methodology as other pair-based datasets.

**Sequence classification datasets:**

- **XNLI** (Conneau et al., 2018) — Natural language inference task extending MNLI (Williams et al., 2018) to non-English languages, consisting in classifying sentence pairs into entailment, contradiction, or neutral.

- **PAWS-X** (Yang et al., 2019) — Paraphrase identification task aimed at determining whether two sentences convey the same meaning. Fine-tuning is performed cross-lingually, with training on the English subset and evaluation across all available languages.

- **AmazonReviews** (Keung et al., 2020) — Sentiment analysis task consisting in estimating the satisfaction level of multilingual Amazon product reviews on a 1-to-5 scale. Fine-tuning is performed on all available languages.

- **MassiveIntent** (Keung et al., 2020) — Multilingual classification task consisting in assigning sentences to one of 60 topic categories. Fine-tuning is performed on all available languages.

- **CodeDefect** (Zhou et al., 2019) — Binary classification task aimed at identifying whether a given code snippet contains a defect.

- **CodeComplexity** (Jeon et al., 2023) — Computational analysis task consisting in estimating the order of complexity of a code-formulated computer science problem.

- **MathShepherd** (Wang et al., 2024b) — Binary classification task aimed at determining whether a step-by-step math rationale is correct given a problem prompt. We limited the dataset to rationales with 3 steps to mitigate the class imbalance observed in longer rationales, where incorrect solutions become more frequent. As the dataset lacks a validation split, we allocate half of the test set for validation.

**Sequence regression datasets:**

- **WMT** (Bojar et al., 2017; 2018; Barrault et al., 2019; 2020; Akhbardeh et al., 2021; Kocmi et al., 2022) — Regression task consisting in estimating translation quality given a source sentence, and possibly a reference translation. As the original test set covers only three language pairs, we construct validation and test sets by sampling 5% of the training set for each, ensuring broader language coverage in evaluation. We report results under both the reference-free and reference-based evaluation settings.

- **SeaHorse** (Clark et al., 2023) — Multilingual summarization evaluation task, where each text-summary pair is annotated across 6 binary evaluation dimensions. The final score is obtained by averaging these labels, yielding a continuous value between 0 and 1. To avoid penalizing models with limited context lengths, the summary is placed first in the input, followed by the main text, ensuring the model can attend to the full summary.

**Token classification datasets:**

- **NER** (Liang et al., 2020) — Named entity recognition task from the XGLUE benchmark, combining subsets of CoNLL2002 (Tjong Kim Sang, 2002) and CoNLL2003 (Tjong Kim Sang & De Meulder, 2003), adapted for cross-lingual evaluation with English-only training. It spans four languages (English, German, Spanish, and Dutch) and targets four entity types (Person, Location, Organization, and Miscellaneous).

# E  Results on larger hyper-parameter search

In this appendix, we investigate why the largest EuroBERT model does not significantly outperform its 610M counterpart on retrieval tasks. We hypothesize the larger model may benefit from a more comprehensive hyper-parameter search. To test this hypothesis, we perform a more extensive grid search for fine-tuning on the MS-MARCO dataset across the EuroBERT family. Rather than varying only the learning rate with fixed optimization settings, we systematically explore a broader set of hyperparameters: Adam's $\beta_2$ (0.95 [default], 0.98, 0.999), Adam's $\epsilon$ ($10^{-5}$ [default], $10^{-8}$), and the number of training steps (1,000 [default], 2,000).

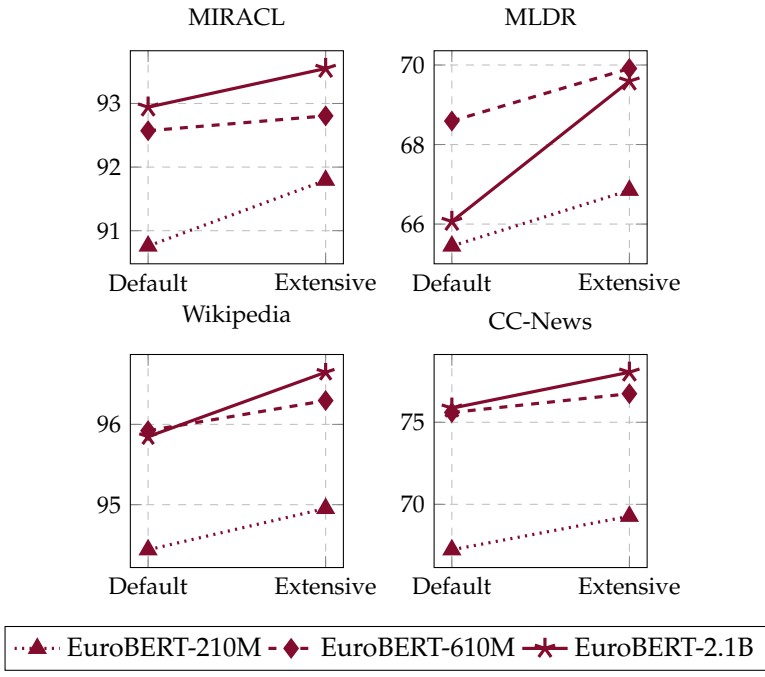

Figure 8: **Retrieval performance** of EuroBERT models under the default fine-tuning configuration (Default) compared to the more extensive hyperparameter grid search (Extensive). Results are reported as average nDCG@10 across supported languages.

Figure 8 demonstrates that, while all models benefit from a denser hyperparameter grid search, the largest EuroBERT model exhibits the most substantial improvements, particularly on the MIRACL and MLDR datasets. Additionally, as shown in Table 8, increasing the number of training steps from 1,000 to 2,000 consistently enhances performance across all model sizes. Also, we observe that models generally benefit form increasing $\beta_2$ and reducing the $\epsilon$ and learning rate.

| | MIRACL | | | | | | | |
|---|---|---|---|---|---|---|---|---|
| | **Learning Rate** | | **Adam $\beta_2$** | | **Adam $\epsilon$** | | **Steps** | |
| **Model** | Default | Extensive | Default | Extensive | Default | Extensive | Default | Extensive |
| EuroBERT-210M | 4.6e-05 | 2.8e-05 | 0.95 | 0.98 | 1e-05 | 1e-05 | 1,000 | 2,000 |
| EuroBERT-610M | 3.6e-05 | 2.8e-05 | 0.95 | 0.98 | 1e-05 | 1e-08 | 1,000 | 1,000 |
| EuroBERT-2.1B | 3.6e-05 | 1.7e-05 | 0.95 | 0.98 | 1e-05 | 1e-08 | 1,000 | 2,000 |
| | MLDR | | | | | | | |
| | **Learning Rate** | | **Adam $\beta_2$** | | **Adam $\epsilon$** | | **Steps** | |
| **Model** | Default | Extensive | Default | Extensive | Default | Extensive | Default | Extensive |
| EuroBERT-210M | 2.8e-05 | 2.8e-05 | 0.95 | 0.98 | 1e-05 | 1e-05 | 1,000 | 2,000 |
| EuroBERT-610M | 2.2e-05 | 2.8e-05 | 0.95 | 0.95 | 1e-05 | 1e-05 | 1,000 | 2,000 |
| EuroBERT-2.1B | 4.6e-05 | 1.3e-05 | 0.95 | 0.98 | 1e-05 | 1e-08 | 1,000 | 2,000 |
| | Wikipedia | | | | | | | |
| | **Learning Rate** | | **Adam $\beta_2$** | | **Adam $\epsilon$** | | **Steps** | |
| **Model** | Default | Extensive | Default | Extensive | Default | Extensive | Default | Extensive |
| EuroBERT-210M | 2.8e-05 | 2.2e-05 | 0.95 | 0.98 | 1e-05 | 1e-08 | 1,000 | 2,000 |
| EuroBERT-610M | 3.6e-05 | 2.2e-05 | 0.95 | 0.95 | 1e-05 | 1e-08 | 1,000 | 2,000 |
| EuroBERT-2.1B | 2.8e-05 | 2.8e-05 | 0.95 | 0.95 | 1e-05 | 1e-05 | 1,000 | 2,000 |
| | CC-News | | | | | | | |
| | **Learning Rate** | | **Adam $\beta_2$** | | **Adam $\epsilon$** | | **Steps** | |
| **Model** | Default | Extensive | Default | Extensive | Default | Extensive | Default | Extensive |
| EuroBERT-210M | 4.6e-05 | 3.6e-05 | 0.95 | 0.98 | 1e-05 | 1e-05 | 1,000 | 2,000 |
| EuroBERT-610M | 4.6e-05 | 2.8e-05 | 0.95 | 0.95 | 1e-05 | 1e-05 | 1,000 | 2,000 |
| EuroBERT-2.1B | 3.6e-05 | 3.6e-05 | 0.95 | 0.95 | 1e-05 | 1e-05 | 1,000 | 2,000 |

Table 8: Overview of the optimal hyperparameters selected on the validation set for both the Default and Extensive fine-tuning configurations.

# F   Detailed Results

Table 9 and Table 10 present per-language results for the retrieval and sequence classification tasks, respectively. Table 11 and Table 12 report detailed performance on sequence regression. Finally, Table 13 provides per-language results on token classification (NER task).

| | MIRACL | | | | | | | | | | | | | | | | |
|---|---|---|---|---|---|---|---|---|---|---|---|---|---|---|---|---|---|
| | European Languages | | | | | | | | Extra-European Languages | | | | | | | Average | |
| Model | en | de | es | fr | it | nl | pl | pt | ar | hi | ja | ru | tr | vi | zh | Euro | World |
| XLM-RoBERTa-280M | 88.0 | — | 88.6 | 91.9 | — | — | — | — | 84.6 | 77.9 | 85.7 | 82.5 | — | — | 83.8 | 89.5 | 85.4 |
| XLM-RoBERTa-560M | 89.7 | — | 91.6 | 93.5 | — | — | — | — | 89.0 | 81.1 | 91.1 | 88.6 | — | — | 90.3 | 91.6 | 89.4 |
| XLM-RoBERTa-3.5B | 90.4 | — | 93.0 | 94.5 | — | — | — | — | 91.5 | 85.1 | 92.6 | 92.1 | — | — | 91.9 | 92.6 | 91.4 |
| mDeBERTa-v3-280M | 45.3 | — | 39.6 | 46.2 | — | — | — | — | 34.4 | 36.0 | 33.7 | 29.6 | — | — | 35.3 | 43.7 | 37.5 |
| mGTE-MLM-305M | 91.4 | — | 94.6 | 95.2 | — | — | — | — | 91.6 | 85.3 | 91.5 | 88.3 | — | — | 91.4 | 93.8 | 91.2 |
| ModernBERT-150M | 93.4 | — | 71.2 | 81.2 | — | — | — | — | 8.6 | 3.0 | 36.0 | 21.3 | — | — | 45.6 | 81.9 | 45.0 |
| ModernBERT-395M | 95.1 | — | 84.2 | 90.7 | — | — | — | — | 5.7 | 8.7 | 32.7 | 19.2 | — | — | 34.3 | 90.0 | 46.3 |
| **EuroBERT-210M** | 94.1 | — | 95.4 | 95.8 | — | — | — | — | 90.0 | 83.2 | 90.8 | 85.7 | — | — | 90.9 | 95.1 | 90.8 |
| **EuroBERT-610M** | 93.6 | — | 95.1 | 96.3 | — | — | — | — | 91.8 | 88.3 | 92.4 | 90.7 | — | — | 92.4 | 95.0 | 92.6 |
| **EuroBERT-2.1B** | 94.2 | — | 95.0 | 95.3 | — | — | — | — | 93.0 | 87.1 | 93.4 | 91.5 | — | — | 94.1 | 94.8 | 92.9 |

| | MLDR | | | | | | | | | | | | | | | | |
|---|---|---|---|---|---|---|---|---|---|---|---|---|---|---|---|---|---|
| | European Languages | | | | | | | | Extra-European Languages | | | | | | | Average | |
| Model | en | de | es | fr | it | nl | pl | pt | ar | hi | ja | ru | tr | vi | zh | Euro | World |
| XLM-RoBERTa-280M | 59.4 | 56.7 | 58.0 | 64.5 | 53.4 | — | — | 60.3 | 44.4 | 56.4 | 50.3 | 43.6 | — | — | 53.8 | 58.7 | 54.6 |
| XLM-RoBERTa-560M | 63.4 | 61.1 | 66.9 | 71.1 | 61.9 | — | — | 67.1 | 51.5 | 60.3 | 54.9 | 51.5 | — | — | 58.9 | 65.2 | 60.8 |
| XLM-RoBERTa-3.5B | 68.9 | 66.1 | 72.7 | 73.5 | 67.5 | — | — | 71.0 | 56.5 | 61.8 | 62.6 | 60.8 | — | — | 63.5 | 70.0 | 65.9 |
| mDeBERTa-v3-280M | 18.8 | 24.3 | 15.4 | 23.9 | 18.2 | — | — | 19.0 | 12.3 | 20.5 | 17.4 | 13.2 | — | — | 18.1 | 20.0 | 18.3 |
| mGTE-MLM-305M | 63.5 | 68.7 | 79.5 | 78.2 | 71.4 | — | — | 78.1 | 55.7 | 66.2 | 62.4 | 60.8 | — | — | 60.9 | 73.2 | 67.8 |
| ModernBERT-150M | 61.0 | 11.3 | 23.5 | 25.8 | 18.4 | — | — | 19.7 | 0.7 | 0.7 | 2.9 | 2.5 | — | — | 0.8 | 26.6 | 15.2 |
| ModernBERT-395M | 68.4 | 25.3 | 42.6 | 58.0 | 21.2 | — | — | 37.1 | 0.5 | 2.0 | 3.5 | 3.7 | — | — | 0.9 | 42.1 | 23.9 |
| **EuroBERT-210M** | 67.2 | 68.1 | 78.2 | 80.0 | 68.9 | — | — | 77.9 | 52.1 | 51.3 | 60.8 | 59.1 | — | — | 56.4 | 73.4 | 65.4 |
| **EuroBERT-610M** | 72.5 | 69.5 | 80.3 | 79.8 | 73.9 | — | — | 79.0 | 55.5 | 60.9 | 61.6 | 62.5 | — | — | 59.0 | 75.8 | 68.6 |
| **EuroBERT-2.1B** | 72.5 | 65.4 | 77.6 | 77.6 | 69.2 | — | — | 75.0 | 53.0 | 58.1 | 61.5 | 59.3 | — | — | 57.6 | 72.9 | 66.1 |

| | Wikipedia | | | | | | | | | | | | | | | | |
|---|---|---|---|---|---|---|---|---|---|---|---|---|---|---|---|---|---|
| | European Languages | | | | | | | | Extra-European Languages | | | | | | | Average | |
| Model | en | de | es | fr | it | nl | pl | pt | ar | hi | ja | ru | tr | vi | zh | Euro | World |
| XLM-RoBERTa-280M | 94.7 | 91.2 | — | — | 91.5 | 90.2 | — | 90.8 | — | 87.4 | — | — | — | — | — | 91.7 | 91.0 |
| XLM-RoBERTa-560M | 95.4 | 93.5 | — | — | 93.4 | 93.4 | — | 92.5 | — | 90.7 | — | — | — | — | — | 93.6 | 93.1 |
| XLM-RoBERTa-3.5B | 97.9 | 96.5 | — | — | 96.6 | 96.3 | — | 96.0 | — | 94.5 | — | — | — | — | — | 96.7 | 96.3 |
| mDeBERTa-v3-280M | 66.1 | 60.4 | — | — | 53.6 | 57.6 | — | 56.5 | — | 51.3 | — | — | — | — | — | 58.9 | 57.6 |
| mGTE-MLM-305M | 96.7 | 93.9 | — | — | 94.7 | 93.9 | — | 93.6 | — | 92.0 | — | — | — | — | — | 94.6 | 94.1 |
| ModernBERT-150M | 97.3 | 56.5 | — | — | 60.5 | 57.2 | — | 67.0 | — | 5.7 | — | — | — | — | — | 67.7 | 57.4 |
| ModernBERT-395M | 98.2 | 69.9 | — | — | 70.0 | 67.4 | — | 83.2 | — | 12.0 | — | — | — | — | — | 77.7 | 66.8 |
| **EuroBERT-210M** | 97.7 | 94.7 | — | — | 94.5 | 95.6 | — | 95.7 | — | 88.6 | — | — | — | — | — | 95.6 | 94.4 |
| **EuroBERT-610M** | 98.3 | 95.9 | — | — | 96.0 | 96.6 | — | 96.1 | — | 92.6 | — | — | — | — | — | 96.6 | 95.9 |
| **EuroBERT-2.1B** | 99.0 | 96.1 | — | — | 96.0 | 96.0 | — | 95.9 | — | 92.0 | — | — | — | — | — | 96.6 | 95.8 |

| | CC-News | | | | | | | | | | | | | | | | |
|---|---|---|---|---|---|---|---|---|---|---|---|---|---|---|---|---|---|
| | European Languages | | | | | | | | Extra-European Languages | | | | | | | Average | |
| Model | en | de | es | fr | it | nl | pl | pt | ar | hi | ja | ru | tr | vi | zh | Euro | World |
| XLM-RoBERTa-280M | 69.9 | 57.8 | 55.8 | 57.5 | 57.2 | 66.8 | 55.1 | 63.1 | 72.2 | 31.1 | 75.7 | 76.2 | 47.8 | 61.5 | 77.1 | 60.4 | 61.6 |
| XLM-RoBERTa-560M | 77.3 | 68.6 | 69.1 | 70.1 | 70.8 | 75.9 | 69.6 | 75.3 | 82.8 | 51.0 | 82.3 | 83.3 | 61.1 | 73.7 | 81.2 | 72.1 | 72.8 |
| XLM-RoBERTa-3.5B | 84.4 | 77.4 | 79.7 | 79.1 | 79.6 | 83.8 | 79.5 | 84.0 | 88.1 | 59.5 | 87.4 | 88.8 | 72.1 | 82.9 | 86.8 | 80.9 | 80.9 |
| mDeBERTa-v3-280M | 25.0 | 15.7 | 11.9 | 12.4 | 13.1 | 20.4 | 12.4 | 15.8 | 23.1 | 4.7 | 33.6 | 23.6 | 10.7 | 20.4 | 34.8 | 15.8 | 18.5 |
| mGTE-MLM-305M | 76.1 | 68.7 | 72.8 | 70.1 | 68.4 | 76.5 | 65.1 | 74.3 | 79.6 | 32.5 | 85.1 | 83.3 | 56.7 | 72.3 | 88.2 | 71.5 | 71.3 |
| ModernBERT-150M | 75.6 | 16.1 | 15.6 | 14.4 | 15.0 | 29.6 | 7.0 | 10.9 | 2.3 | 1.8 | 6.7 | 2.5 | 8.5 | 5.2 | 10.0 | 23.0 | 14.7 |
| ModernBERT-395M | 84.9 | 21.5 | 33.4 | 41.3 | 20.0 | 36.2 | 4.2 | 27.8 | 2.2 | 2.0 | 9.5 | 4.1 | 9.6 | 10.0 | 9.9 | 33.6 | 21.1 |
| **EuroBERT-210M** | 80.0 | 66.9 | 69.2 | 69.7 | 65.9 | 73.5 | 57.8 | 69.1 | 76.7 | 17.0 | 82.4 | 79.7 | 52.1 | 57.4 | 90.9 | 69.0 | 67.2 |
| **EuroBERT-610M** | 84.0 | 72.9 | 76.4 | 75.5 | 73.8 | 79.6 | 70.9 | 79.9 | 84.0 | 49.7 | 84.9 | 85.1 | 62.4 | 66.7 | 88.1 | 76.6 | 75.6 |
| **EuroBERT-2.1B** | 85.8 | 73.1 | 77.1 | 76.9 | 73.8 | 79.0 | 70.3 | 79.7 | 84.2 | 49.9 | 88.2 | 86.5 | 60.7 | 63.0 | 89.9 | 76.9 | 75.9 |

Table 9: Detailed results on multilingual retrieval tasks (nDCG@10, in %).

**XNLI**

| Model | European Languages | | | | | | | | Extra-European Languages | | | | | | | Average | |
|---|---|---|---|---|---|---|---|---|---|---|---|---|---|---|---|---|---|
| | en | de | es | fr | it | nl | pl | pt | ar | hi | ja | ru | tr | vi | zh | Euro | World |
| XLM-RoBERTa-280M | 80.0 | 74.8 | 76.4 | 75.1 | — | — | — | — | 71.5 | 69.1 | — | 74.4 | 72.4 | 74.0 | 73.2 | 76.6 | 74.1 |
| XLM-RoBERTa-560M | 87.1 | 83.0 | 83.5 | 82.9 | — | — | — | — | 80.4 | 77.8 | — | 81.2 | 80.6 | 80.2 | 80.6 | 84.1 | 81.7 |
| XLM-RoBERTa-3.5B | 89.0 | 85.5 | 85.3 | 84.5 | — | — | — | — | 81.5 | 80.5 | — | 83.1 | 82.4 | 83.3 | 82.3 | 86.1 | 83.7 |
| mDeBERTa-v3-280M | 84.9 | 81.2 | 81.1 | 81.0 | — | — | — | — | 77.6 | 76.1 | — | 79.1 | 77.7 | 78.1 | 78.4 | 82.0 | 79.5 |
| mGTE-MLM-305M | 81.1 | 76.9 | 78.5 | 77.2 | — | — | — | — | 73.6 | 71.3 | — | 75.4 | 72.9 | 75.9 | 75.5 | 78.4 | 75.8 |
| ModernBERT-150M | 82.8 | 65.7 | 69.9 | 70.6 | — | — | — | — | 56.9 | 54.0 | — | 63.1 | 53.9 | 58.3 | 68.1 | 72.3 | 64.3 |
| ModernBERT-395M | 89.4 | 75.6 | 79.2 | 79.1 | — | — | — | — | 59.6 | 55.7 | — | 70.9 | 60.6 | 63.0 | 76.1 | 80.8 | 70.9 |
| **EuroBERT-210M** | 83.5 | 77.8 | 79.4 | 78.9 | — | — | — | — | 74.3 | 70.6 | — | 76.6 | 74.2 | 75.1 | 75.3 | 79.9 | 76.6 |
| **EuroBERT-610M** | 87.8 | 82.9 | 84.6 | 83.6 | — | — | — | — | 79.5 | 76.7 | — | 82.0 | 80.3 | 80.8 | 80.7 | 84.7 | 81.9 |
| **EuroBERT-2.1B** | 89.6 | 85.5 | 86.4 | 85.8 | — | — | — | — | 82.8 | 79.9 | — | 83.3 | 83.0 | 82.3 | 82.3 | 86.8 | 84.1 |

**PAWS-X**

| Model | European Languages | | | | | | | | Extra-European Languages | | | | | | | Average | |
|---|---|---|---|---|---|---|---|---|---|---|---|---|---|---|---|---|---|
| | en | de | es | fr | it | nl | pl | pt | ar | hi | ja | ru | tr | vi | zh | Euro | World |
| XLM-RoBERTa-280M | 93.8 | 86.4 | 87.5 | 88.0 | — | — | — | — | — | — | — | — | — | — | — | 88.9 | 88.9 |
| XLM-RoBERTa-560M | 95.5 | 91.0 | 91.4 | 91.8 | — | — | — | — | — | — | — | — | — | — | — | 92.4 | 92.4 |
| XLM-RoBERTa-3.5B | 95.8 | 91.9 | 91.7 | 92.3 | — | — | — | — | — | — | — | — | — | — | — | 92.9 | 92.9 |
| mDeBERTa-v3-280M | 95.7 | 90.2 | 90.4 | 91.3 | — | — | — | — | — | — | — | — | — | — | — | 91.9 | 91.9 |
| mGTE-MLM-305M | 94.7 | 87.5 | 88.2 | 88.8 | — | — | — | — | — | — | — | — | — | — | — | 89.8 | 89.8 |
| ModernBERT-150M | 94.7 | 72.0 | 74.0 | 77.7 | — | — | — | — | — | — | — | — | — | — | — | 79.6 | 79.6 |
| ModernBERT-395M | 95.8 | 75.4 | 82.7 | 83.5 | — | — | — | — | — | — | — | — | — | — | — | 84.3 | 84.3 |
| **EuroBERT-210M** | 95.6 | 86.5 | 88.7 | 88.9 | — | — | — | — | — | — | — | — | — | — | — | 89.9 | 89.9 |
| **EuroBERT-610M** | 95.6 | 90.0 | 91.3 | 92.0 | — | — | — | — | — | — | — | — | — | — | — | 92.2 | 92.2 |
| **EuroBERT-2.1B** | 96.2 | 91.6 | 91.8 | 92.5 | — | — | — | — | — | — | — | — | — | — | — | 93.0 | 93.0 |

**AmazonReviews**

| Model | European Languages | | | | | | | | Extra-European Languages | | | | | | | Average | |
|---|---|---|---|---|---|---|---|---|---|---|---|---|---|---|---|---|---|
| | en | de | es | fr | it | nl | pl | pt | ar | hi | ja | ru | tr | vi | zh | Euro | World |
| XLM-RoBERTa-280M | 64.9 | 65.0 | 60.6 | 60.0 | — | — | — | — | — | — | 59.0 | — | — | — | 56.8 | 62.7 | 61.1 |
| XLM-RoBERTa-560M | 66.9 | 67.0 | 62.4 | 61.5 | — | — | — | — | — | — | 62.1 | — | — | — | 58.5 | 64.5 | 63.1 |
| XLM-RoBERTa-3.5B | 67.1 | 67.8 | 62.4 | 61.5 | — | — | — | — | — | — | 63.7 | — | — | — | 59.2 | 63.7 | 63.6 |
| mDeBERTa-v3-280M | 66.4 | 66.1 | 61.6 | 60.6 | — | — | — | — | — | — | 60.4 | — | — | — | 57.7 | 63.7 | 62.1 |
| mGTE-MLM-305M | 65.0 | 65.3 | 60.9 | 59.5 | — | — | — | — | — | — | 61.2 | — | — | — | 57.4 | 62.7 | 61.5 |
| ModernBERT-150M | 66.1 | 61.4 | 57.5 | 57.8 | — | — | — | — | — | — | 54.2 | — | — | — | 53.9 | 60.7 | 58.5 |
| ModernBERT-395M | 67.6 | 64.9 | 60.8 | 60.0 | — | — | — | — | — | — | 58.2 | — | — | — | 57.8 | 64.3 | 61.5 |
| **EuroBERT-210M** | 65.9 | 65.4 | 60.5 | 60.2 | — | — | — | — | — | — | 60.4 | — | — | — | 57.7 | 63.0 | 61.7 |
| **EuroBERT-610M** | 66.7 | 66.4 | 61.6 | 61.2 | — | — | — | — | — | — | 61.7 | — | — | — | 58.1 | 64.0 | 62.6 |
| **EuroBERT-2.1B** | 66.5 | 67.8 | 62.8 | 60.9 | — | — | — | — | — | — | 62.4 | — | — | — | 59.0 | 64.5 | 63.2 |

**MassiveIntent**

| Model | European Languages | | | | | | | | Extra-European Languages | | | | | | | Average | |
|---|---|---|---|---|---|---|---|---|---|---|---|---|---|---|---|---|---|
| | en | de | es | fr | it | nl | pl | pt | ar | hi | ja | ru | tr | vi | zh | Euro | World |
| XLM-RoBERTa-280M | 89.1 | 86.1 | 87.0 | 86.6 | 87.5 | 87.5 | 86.8 | 87.3 | 79.0 | 86.2 | 86.5 | 87.3 | 85.6 | 86.7 | 86.0 | 87.2 | 86.3 |
| XLM-RoBERTa-560M | 90.3 | 87.7 | 88.0 | 89.0 | 88.5 | 89.1 | 88.8 | 88.8 | 83.5 | 88.1 | 88.9 | 89.0 | 87.8 | 88.9 | 87.1 | 88.8 | 88.2 |
| XLM-RoBERTa-3.5B | 89.9 | 87.6 | 88.2 | 88.6 | 88.7 | 88.3 | 87.9 | 88.6 | 81.8 | 88.0 | 88.4 | 89.2 | 87.8 | 88.4 | 86.9 | 88.5 | 87.9 |
| mDeBERTa-v3-280M | 88.1 | 86.4 | 86.9 | 87.3 | 87.6 | 88.0 | 87.0 | 86.9 | 79.8 | 86.0 | 87.3 | 87.3 | 85.9 | 86.5 | 85.9 | 87.3 | 86.5 |
| mGTE-MLM-305M | 89.0 | 86.3 | 87.4 | 87.9 | 87.2 | 87.9 | 86.3 | 87.8 | 80.7 | 86.6 | 87.9 | 87.8 | 86.4 | 87.4 | 86.3 | 87.5 | 86.9 |
| ModernBERT-150M | 85.5 | 74.8 | 76.6 | 79.4 | 75.1 | 74.2 | 71.1 | 78.2 | 57.6 | 56.1 | 76.6 | 74.4 | 62.3 | 66.3 | 79.9 | 76.9 | 72.6 |
| ModernBERT-395M | 89.8 | 83.9 | 84.9 | 86.9 | 84.7 | 84.1 | 82.4 | 86.0 | 72.5 | 76.7 | 84.4 | 83.9 | 79.6 | 80.8 | 84.9 | 85.3 | 83.0 |
| **EuroBERT-210M** | 89.0 | 86.0 | 86.9 | 86.9 | 87.0 | 87.1 | 86.8 | 87.9 | 81.2 | 86.9 | 87.4 | 87.2 | 85.8 | 86.0 | 86.0 | 87.2 | 86.5 |
| **EuroBERT-610M** | 89.2 | 86.6 | 87.4 | 87.6 | 88.1 | 88.2 | 87.3 | 87.8 | 82.7 | 87.3 | 88.3 | 88.2 | 86.8 | 86.1 | 87.0 | 87.8 | 87.2 |
| **EuroBERT-2.1B** | 88.9 | 87.2 | 88.0 | 88.7 | 87.9 | 88.2 | 88.1 | 88.2 | 83.2 | 87.6 | 89.0 | 88.1 | 87.1 | 85.4 | 87.0 | 88.2 | 87.5 |

Table 10: Detailed results on multilingual sequence classification tasks (accuracy, in %).

| | Ref-free | | | | | | | | | | | | | | Average | |
| | European Pairs | | | | | | Extra-European Pairs | | | | | | | | | |
| | en-xx | | xx-en | | Other | | en-xx | | | | xx-en | | | | Euro | World |
| Model | en-de | en-pl | de-en | pl-en | de-fr | fr-de | en-ja | en-ru | en-tr | en-zh | ja-en | ru-en | tr-en | zh-en | | |
| XLM-RoBERTa-280M | 45.3 | 53.6 | 26.7 | 16.3 | 31.4 | 32.0 | 47.0 | 56.5 | 61.5 | 42.5 | 10.4 | 21.8 | 40.7 | 25.3 | 34.2 | 36.5 |
| XLM-RoBERTa-560M | 50.7 | 66.0 | 30.7 | 15.8 | 41.1 | 29.8 | 52.1 | 61.2 | 66.2 | 47.2 | 11.0 | 24.8 | 45.4 | 29.0 | 39.0 | 40.8 |
| XLM-RoBERTa-3.5B | 55.1 | 66.9 | 35.9 | 18.7 | 46.7 | 43.3 | 56.3 | 64.9 | 64.8 | 52.2 | 13.1 | 27.0 | 47.7 | 30.6 | 44.4 | 44.5 |
| mDeBERTa-v3-280M | 52.8 | 61.6 | 33.1 | 18.3 | 44.4 | 39.6 | 51.6 | 61.2 | 67.2 | 46.7 | 11.2 | 24.2 | 47.3 | 28.9 | 41.6 | 42.0 |
| mGTE-MLM-305M | 48.6 | 55.2 | 30.4 | 18.5 | 37.5 | 35.9 | 48.3 | 57.2 | 59.7 | 45.5 | 10.6 | 23.4 | 41.5 | 27.3 | 37.7 | 38.5 |
| ModernBERT-150M | 39.7 | 47.4 | 29.8 | 18.2 | 20.5 | 21.9 | 36.0 | 41.6 | 39.6 | 37.8 | 10.7 | 21.5 | 41.3 | 23.9 | 29.6 | 30.7 |
| ModernBERT-395M | 45.3 | 51.7 | 32.4 | 20.0 | 23.8 | 27.6 | 37.8 | 44.7 | 43.6 | 41.5 | 12.3 | 24.1 | 42.7 | 27.1 | 33.5 | 33.9 |
| **EuroBERT-210M** | 52.9 | 58.4 | 33.2 | 17.5 | 40.6 | 40.3 | 51.1 | 57.9 | 57.3 | 48.3 | 14.3 | 26.7 | 44.3 | 30.8 | 40.5 | 41.0 |
| **EuroBERT-610M** | 52.9 | 61.1 | 32.4 | 18.2 | 42.6 | 39.2 | 51.3 | 59.4 | 62.3 | 48.6 | 12.2 | 26.6 | 44.1 | 29.7 | 41.1 | 41.5 |
| **EuroBERT-2.1B** | 49.1 | 57.8 | 29.8 | 19.3 | 38.3 | 38.5 | 47.8 | 56.9 | 56.5 | 45.0 | 10.7 | 23.5 | 41.3 | 27.5 | 38.8 | 38.7 |
| | Ref-based | | | | | | | | | | | | | | Average | |
| | European Pairs | | | | | | Extra-European Pairs | | | | | | | | | |
| | en-xx | | xx-en | | Other | | en-xx | | | | xx-en | | | | Euro | World |
| Model | en-de | en-pl | de-en | pl-en | de-fr | fr-de | en-ja | en-ru | en-tr | en-zh | ja-en | ru-en | tr-en | zh-en | | |
| XLM-RoBERTa-280M | 49.6 | 56.0 | 34.4 | 29.2 | 47.8 | 41.9 | 49.8 | 60.5 | 63.1 | 50.1 | 14.2 | 25.3 | 48.7 | 31.1 | 43.1 | 43.0 |
| XLM-RoBERTa-560M | 52.3 | 63.9 | 37.3 | 26.8 | 51.1 | 42.3 | 53.6 | 63.3 | 70.9 | 53.0 | 12.6 | 25.2 | 49.2 | 32.3 | 45.6 | 45.3 |
| XLM-RoBERTa-3.5B | 55.9 | 68.2 | 39.1 | 28.2 | 53.7 | 45.6 | 56.7 | 66.3 | 71.1 | 55.2 | 12.8 | 28.9 | 52.4 | 34.0 | 48.5 | 47.7 |
| mDeBERTa-v3-280M | 53.0 | 65.6 | 37.6 | 27.9 | 50.9 | 43.7 | 52.7 | 63.5 | 69.1 | 53.0 | 13.1 | 27.8 | 49.9 | 32.5 | 46.5 | 45.7 |
| mGTE-MLM-305M | 51.0 | 57.4 | 36.6 | 29.9 | 49.6 | 39.3 | 51.5 | 61.8 | 63.5 | 52.4 | 13.8 | 26.0 | 48.7 | 32.8 | 44.0 | 43.9 |
| ModernBERT-150M | 43.5 | 50.2 | 36.8 | 28.2 | 42.5 | 33.1 | 45.4 | 50.4 | 54.2 | 46.0 | 13.8 | 26.6 | 50.8 | 31.7 | 39.1 | 39.5 |
| ModernBERT-395M | 47.4 | 53.9 | 39.2 | 31.6 | 43.5 | 34.8 | 47.1 | 52.3 | 60.0 | 48.6 | 14.9 | 28.9 | 53.1 | 34.0 | 41.7 | 42.1 |
| **EuroBERT-210M** | 52.5 | 59.8 | 38.8 | 30.5 | 49.7 | 39.6 | 52.2 | 61.5 | 65.9 | 53.1 | 15.2 | 28.7 | 50.5 | 34.7 | 45.1 | 45.2 |
| **EuroBERT-610M** | 53.8 | 64.0 | 37.8 | 29.4 | 51.3 | 42.5 | 53.2 | 62.9 | 67.9 | 52.3 | 16.2 | 29.2 | 50.6 | 33.5 | 46.5 | 46.0 |
| **EuroBERT-2.1B** | 54.9 | 65.7 | 39.4 | 31.0 | 54.9 | 45.0 | 55.0 | 64.3 | 68.3 | 54.5 | 14.9 | 28.3 | 52.0 | 34.3 | 48.5 | 47.3 |

Table 11: Detailed results on the WMT tasks (Spearman rank correlation, in %).

| | European Languages | | | | | | | | Extra-European Languages | | | | | | | | Average | |
| Model | en | de | es | fr | it | nl | pl | pt | ar | hi | ja | ru | tr | vi | zh | | Euro | World |
| XLM-RoBERTa-280M | 53.2 | 55.6 | 62.0 | — | — | — | — | — | — | — | — | 69.1 | 67.3 | 59.3 | — | | 56.9 | 61.1 |
| XLM-RoBERTa-560M | 57.1 | 60.1 | 66.9 | — | — | — | — | — | — | — | — | 73.6 | 72.6 | 62.7 | — | | 61.4 | 65.5 |
| XLM-RoBERTa-3.5B | 58.1 | 62.1 | 69.7 | — | — | — | — | — | — | — | — | 75.6 | 75.5 | 64.0 | — | | 63.3 | 67.5 |
| mDeBERTa-v3-280M | 56.2 | 58.5 | 66.3 | — | — | — | — | — | — | — | — | 72.6 | 71.9 | 59.6 | — | | 60.3 | 64.2 |
| mGTE-MLM-305M | 52.7 | 58.6 | 66.3 | — | — | — | — | — | — | — | — | 69.7 | 69.6 | 61.4 | — | | 59.2 | 63.0 |
| ModernBERT-150M | 44.6 | 47.9 | 52.8 | — | — | — | — | — | — | — | — | 59.6 | 53.8 | 52.1 | — | | 48.4 | 51.8 |
| ModernBERT-395M | 56.9 | 56.8 | 64.0 | — | — | — | — | — | — | — | — | 66.7 | 65.6 | 59.0 | — | | 59.3 | 61.5 |
| **EuroBERT-210M** | 54.4 | 58.7 | 67.2 | — | — | — | — | — | — | — | — | 70.0 | 71.2 | 61.3 | — | | 60.1 | 63.8 |
| **EuroBERT-610M** | 57.2 | 60.6 | 70.3 | — | — | — | — | — | — | — | — | 72.6 | 73.5 | 61.5 | — | | 62.7 | 66.0 |
| **EuroBERT-2.1B** | 58.8 | 62.1 | 71.1 | — | — | — | — | — | — | — | — | 74.4 | 75.7 | 62.8 | — | | 64.0 | 67.5 |

Table 12: Detailed results on the SeaHorse summary evaluation task (Spearman rank correlation, in %).

| | European Languages | | | | | | | | Extra-European Languages | | | | | | | | Average | |
| Model | en | de | es | fr | it | nl | pl | pt | ar | hi | ja | ru | tr | vi | zh | | Euro | World |
| XLM-RoBERTa-280M | 97.6 | 94.5 | 93.6 | — | — | 96.1 | — | — | — | — | — | — | — | — | — | | 95.5 | 95.5 |
| XLM-RoBERTa-560M | 97.7 | 96.4 | 93.9 | — | — | 96.2 | — | — | — | — | — | — | — | — | — | | 96.1 | 96.1 |
| XLM-RoBERTa-3.5B | 98.3 | 96.4 | 94.5 | — | — | 96.1 | — | — | — | — | — | — | — | — | — | | 96.3 | 96.3 |
| mDeBERTa-v3-280M | 98.1 | 96.4 | 94.2 | — | — | 96.1 | — | — | — | — | — | — | — | — | — | | 96.2 | 96.2 |
| mGTE-MLM-305M | 97.9 | 94.0 | 93.0 | — | — | 95.7 | — | — | — | — | — | — | — | — | — | | 95.2 | 95.2 |
| ModernBERT-150M | 84.0 | 89.4 | 89.8 | — | — | 92.8 | — | — | — | — | — | — | — | — | — | | 89.0 | 89.0 |
| ModernBERT-395M | 97.8 | 81.5 | 92.5 | — | — | 94.6 | — | — | — | — | — | — | — | — | — | | 91.6 | 91.6 |
| **EuroBERT-210M** | 97.7 | 91.8 | 93.8 | — | — | 95.5 | — | — | — | — | — | — | — | — | — | | 94.7 | 94.7 |
| **EuroBERT-610M** | 97.6 | 96.0 | 94.0 | — | — | 95.8 | — | — | — | — | — | — | — | — | — | | 95.9 | 95.9 |
| **EuroBERT-2.1B** | 97.6 | 94.2 | 93.8 | — | — | 95.0 | — | — | — | — | — | — | — | — | — | | 95.2 | 95.2 |

Table 13: Detailed results on the NER task (F1 score, in %).

