# OpenReview forum: "EuroBERT: Scaling Multilingual Encoders for European Languages"
_colmweb.org/COLM/2025/Conference — COLM 2025_

### Official Review · Reviewer_6A47 · 2025-05-02

**Rating:** 7
**Confidence:** 4
**Ethics Flag:** 1

**Summary:**

The paper trains new multilingual encoder models of three different sizes (200M, 600M, 2B parameters), adopting various improvements from recent work on decoder-only models, and evaluates them on a range of retrieval, classification and regression benchmarks. They show that the model outperforms or is on par with alternatives across a wide range of tasks and languages. They promise to release the models, intermediate checkpoints and the training code base.

I think this paper is a valuable contribution to the community. The encoders will be useful for a wide range of research on downstream tasks and non-English languages that so far had to rely on outdated encoders. The reported ablation experiments (and promised intermediate checkpoints and training code) will also be very informative for future encoder training. The paper is well-written, easy to follow, and I don't see any major flaws in the experimental setup.

**Reasons To Accept:**

Valuable resource that enables future research. Fills existing gaps (outdated encoders, language coverage). Well documented and released along with code to reproduce.

**Reasons To Reject:**

Minor criticism: The biggest model performs a bit worse than expected, as the authors also acknowledge and investigate in the appendix. A reason could be that all sizes were trained with the same learning rate, batch size and other hyper parameters. Typically, hyper parameters need to scale along with the model size. Existing work even shows how to avoid tuning hyper parameters at larger scale by transferring from small models (see e.g. [1] and [2]). Despite this potential limitation, the trained models are still valuable.

[1] Tensor Programs V: Tuning Large Neural Networks via Zero-Shot Hyperparameter Transfer
[2] Scaling Exponents Across Parameterizations and Optimizers

---

> ### Author Response · Authors · 2025-05-31
>
> Thank you for your positive review! We are glad that you found our work a valuable resource.
>
> > “The biggest model performs a bit worse than expected, as the authors also acknowledge and investigate in the appendix.”
>
> Thank you for your suggestion! We agree it would be interesting to explore hyper-parameter scaling techniques, such as MuP, which could have improved the performance of our largest model. However, given the complexity of pre-training pipelines and our limited budget (which only allowed for a single run), we chose to follow established practices in the literature when choosing our hyper-parameters. We view this aspect as an important area for future work.
>
> We would like to thank the reviewer again for their thoughtful feedback. Please let us know if you have any other questions that we can clarify.

---

> > ### Comment · Reviewer_6A47 · 2025-06-04
> >
> > Thank you very much for the response. I still disagree that training all model sizes with the same hyperparameters is "following established practices in the literature", but as I already wrote, the paper is valuable nevertheless.
> >
> > After reading all reviews and author responses, I still find my original rating to be accurate and will keep it as is.

---

### Official Review · Reviewer_iq5e · 2025-05-12

**Rating:** 6
**Confidence:** 4
**Ethics Flag:** 1

**Summary:**

This paper introduces EuroBERT, a family of multilingual encoder models designed to advance the state-of-the-art in multilingual NLP tasks. The authors address the growing need for robust multilingual encoders by leveraging recent architectural and training advancements from decoder models, while focusing on European languages and global languages. EuroBERT is trained on a 5T-token corpus spanning 15 languages, including code and mathematics, and demonstrates superior performance across retrieval, classification, regression, and token classification tasks compared to existing models like XLM-RoBERTa and mGTE-MLM. The paper also provides an in-depth analysis of design choices, including dataset composition and training strategies, and releases the models, checkpoints, and framework to facilitate future research.

**Reasons To Accept:**

The authors conduct extensive experiments across various tasks, including retrieval (MIRACL, MLDR), classification (XNLI, PAWS-X), regression (WMT, SeaHorse), and token classification (NER). Results demonstrate that EuroBERT consistently outperforms or matches baselines (XLM-RoBERTa, mGTE-MLM) while requiring fewer parameters.

The release of models, checkpoints, and training code is commendable and aligns with reproducibility standards. This transparency will accelerate follow-up work on multilingual encoders.

**Reasons To Reject:**

My major concern of this paper is the innovation seems limited.  The basic idea is to transfer to the training recipe  from decoder models to encoder models, I don't see enough innovation in the proposed method. So maybe the author could elaborate the key new findings of this paper.
And the results seems a little bit empirical and lack in depth explanations, such as in Sec4, Why does instruction tuning degrade encoder performance, contrary to decoders? Is this tied to task objectives or architecture?

---

> ### Author Response · Authors · 2025-05-31
>
> Thank you for your thoughtful review. We appreciate that you value our efforts towards extensive evaluation, reproducibility and transparency.
>
> > “My major concern of this paper is the innovation seems limited.”
>
> We respectfully disagree. Research on pre-training is naturally expensive (pre-training the EuroBERT family required 80 days with 100 GPUs), limiting the breadth of our experiments, but we believe our work contributes with several meaningful insights towards scaling encoder training, both on the training distribution (Figure 4) and hyper-parameter choices (Figure 5). For example, we show that educational data filters used in decoder models lead to degraded performance in encoders (see Figure 5 right). We also show that training on math and code data, a decision where we diverge from previous work, improves multilingual retrieval quality, but degrades multilingual classification (see Figure 4, second and third plots). Overall, our findings highlight several important aspects in training encoders at a large scale, and we believe they are valuable contributions to the research community.
>
> > “And the results seems a little bit empirical and lack in depth explanations”
>
> Indeed, similar to other works in the field, several of our findings are empirical. We conducted ablation experiments and tried to provide explanations/hypotheses for some of them, especially when results diverge from expectations. In many cases, however, finding a full concrete explanation beyond speculation is not possible or would require deeper elaboration worthy of a paper by itself, and to this end, we release all training artifacts (checkpoints, codebase, etc). Overall, our main goal with this paper was to reignite the exploration of encoders, highlighting unexpected behaviours in light of recent advances in decoder models, and proposing effective solutions, which we believe will be valuable to the community.
>
> > “Why does instruction tuning degrade encoder performance, contrary to decoders?”
>
> This is a very interesting point. Similar to the data quality results in our paper, we hypothesise there may be a domain mismatch, as most downstream tasks for encoders are non-conversational. A more promising strategy may be to reserve this data to a later fine-tuning stage, particularly for use cases that may benefit from conversational input, such as evaluating dialogs.
>
> We would like to thank the reviewer again for their feedback. We hope our responses clarified any remaining concerns, and that you'll consider increasing your scores. If you have any remaining questions, we are happy to address them.

---

### Official Review · Reviewer_pgGu · 2025-05-13

**Rating:** 7
**Confidence:** 4
**Ethics Flag:** 1

**Summary:**

This paper presents a large-scale multilingual encoder, trained on European languages as well as a few non European major languages with a masked language modeling objective. The model is evaluated on several multilingual tasks, including code and math related.
Results show that the proposed model is competitive on various tasks and state-of-the-art on code/math tasks.

**Questions To Authors:**

* 98: "due to pre-processing constraints": more details on those constraints would be welcome.
* on non code/math tasks, XLM-Roberta still seems to outperform EuroBERT (7 wins vs 6 wins) but has 3.5B parameters compared to 2.1B parameters for EuroBERT. Could increasing capacity bridge the gap?
* 180-181: "whereas XLM-RoBERTa suffers notable degradation." this seems accurate on MLDR but not on SeaHorse

**Reasons To Accept:**

* the experimental results are thorough and strong
* the model will be publicly released (although I cannot find a hyperlink or supplementary material with the models available yet), which could benefit the community
* the paper is well written and detail oriented

**Reasons To Reject:**

* even though the results are strong, the fact that XLM-Roberta is ranked 1st 7 times while EuroBERT is ranked first 6 times (see Table 1) could prevent adoption. EuroBERT outperforms all other baselines on math/code but this is likely because related data was added to training.
* a comparison on decoder architecture would be informative

---

> ### Author Response · Authors · 2025-05-31
>
> Thank you for your thorough review and valuable feedback. We are grateful that you found our work well written and experimentally thorough, and that you appreciate our efforts towards publicly releasing all artifacts.
>
> > “XLM-Roberta is ranked 1st 7 times while EuroBERT is ranked first 6 times (see Table 1) could prevent adoption”
>
> We understand the concern, but while XLM-RoBERTa-XL ranks first slightly more often in Table 1, EuroBERT-2.1B offers complementary strengths that we believe make it valuable to the community. Notably, EuroBERT-2.1B is smaller (2.1B vs 3.5B parameters), and supports longer contexts, which enables other types of downstream applications. This is exemplified by comparing both models on the MLDR dataset, a long context retrieval dataset, where all EuroBERT models outperform the XLM-RoBERTa family at longer contexts (Figure 3, left). Additionally, in the interest of facilitating the adoption of our models by the community, we release the code for pre-training and fine-tuning the models, significantly lowering the barrier using the EuroBERT family.
>
> > “EuroBERT outperforms all other baselines on math/code but this is likely because related data was added to training.”
>
> Yes, that is correct. Including math and code data during pre-training, a choice where we diverge from previous work, likely explains the improved performance in those domains. Nevertheless, this choice was not made to optimize performance on these domains, but instead to analyse whether there was transfer to other domains, an hypothesis that has been demonstrated for decoder LLMs [1], but not investigated in the context of encoder models. Therefore, we believe this decision is an advantage: it makes our model more versatile, and thus more useful to the community, and allows us to show that these domains are beneficial for multilingual retrieval but not for multilingual classification (see Figure 4).
>
> [1] Aryabumi et al. To Code, or Not To Code? Exploring Impact of Code in Pre-training
>
> > “a comparison on decoder architecture would be informative”
>
> In this work, we chose to align with prevailing approaches in training small encoder models, which traditionally leaned towards bi-directional architectures. In contrast, recent work has also shown some promising results in adapting larger decoder LLMs to become general purpose embedding models. However, these approaches typically start with models of much larger sizes than the EuroBERT family [2,3]. Further investigating how to apply these approaches using smaller model sizes would be an interesting experiment, but we consider it beyond the current scope of this paper.
>
> [2] Wang et al. Improving Text Embeddings with Large Language Models.
>
> [3] Lee et al. NV-Embed: Improved Techniques for Training LLMs as Generalist Embedding Models.
>
> > “due to pre-processing constraints”
>
> Our dataset preprocessing pipeline pre-packs documents into fixed-sized arrays, which makes implementing variable-length training challenging without significant modifications to the already in-use pre-training codebase. Given the significant risks of these changes, and the fact that our budget only allowed for a single full run, we chose to adopt a simpler alternative: splitting sentences after packing, which only involved changing the model’s attention mask.
>
> > “Could increasing capacity bridge the gap?”
>
> We believe so. Our results indicate that increasing model size leads to significant gains, aligning with broader trends in LLMs. However, we also aimed to have an efficient model that could be more easily used by the community. This factor motivated our choice towards a smaller model trained on more data.
>
> > "whereas XLM-RoBERTa suffers notable degradation."
>
> We agree that the degradation is more visible for the MLDR dataset, but we also observed some degradation on SeaHorse. For the quantiles with larger documents, the gap between similarly sized EuroBERT and XLM-RoBERTa models increases, favouring EuroBERT. In the camera ready, we will update this sentence to further clarify the level of degradation for the different tasks.
>
> We would like to thank the reviewer again for their detailed feedback. We hope to have addressed all your concerns and questions, and that you'll consider raising your score. Please let us know if you have any remaining concerns, we are happy to clarify them.

---

> > ### Comment · Reviewer_pgGu · 2025-06-09
> >
> > Thank you for the detailed rebuttal!
> >
> > > Notably, EuroBERT-2.1B is smaller (2.1B vs 3.5B parameters)
> >
> > This is correct but not very significant. In fact, reading the paper, I was wondering why not make the model slightly larger in order to obtain higher performance.
> >
> > > This is exemplified by comparing both models on the MLDR dataset, a long context retrieval dataset, where all EuroBERT models outperform the XLM-RoBERTa family at longer contexts (Figure 3, left).
> >
> > This seems like a good complementary skill.
> >
> > > Additionally, in the interest of facilitating the adoption of our models by the community, we release the code for pre-training and fine-tuning the models, significantly lowering the barrier using the EuroBERT family.
> >
> > This is already the case for XLM-RoBERTa.
> >
> > > but instead to analyse whether there was transfer to other domains
> >
> > Was there an analysis?
> >
> > > Further investigating how to apply these approaches using smaller model sizes would be an interesting experiment, but we consider it beyond the current scope of this paper.
> >
> > My suggestion was do to a comparison to large LLMs. Admittedly, it would be an unfair comparison, but still informative IMO.
> >
> > While I think this is solid experimental work and could potentially add value to the conference and the community, it's likely that the community would be able to use XLM-RobBERTa effectively as starting point (e.g. continued pretraining with extra code/math data, etc.), so I am inclined to keep my score.

---

> > > ### Author Response · Authors · 2025-06-10
> > >
> > > Thank you for your reply!
> > >
> > > > This seems like a good complementary skill.
> > >
> > > We disagree with the characterization that long context is “a good complementary skill.” Many tasks in NLP inherently require contexts that exceed the 512-token limit of XLM-RoBERTa. For example, encoders are increasingly used for large-scale web data classification, and many documents naturally surpass this range. While exploring these applications is beyond the scope of our work, we believe that offering a model with 16 times the context length of XLM-RoBERTa provides substantial practical value.
> > >
> > > > Was there an analysis?
> > >
> > > In Figure 4, we analyse how varying the ratio of math and code data during the annealing phase impacts performance on multilingual tasks. We show that both math and code positively contribute to multilingual retrieval, but degrade performance in sequence classification. We believe this result is valuable, as it may guide the development of future encoder models.
> > >
> > > >  it's likely that the community would be able to use XLM-RobBERTa effectively as starting point
> > >
> > > While it would likely be possible to improve XLM-RoBERTa, we also consider there would be significant challenges to doing so. XLM-RoBERTa uses absolute position encodings, making it hard to increase the context size. And although further pre-training could enhance math and code capabilities, it may come with a degradation of other capabilities, as observed with EuroBERT in sequence classification. Finally, the EuroBERT family also includes 210M and 610M models that outperform similarly-sized models from the XLM-RoBERTa family in a majority of tasks.

---

### Official Review · Reviewer_aExR · 2025-05-20

**Rating:** 6
**Confidence:** 4
**Ethics Flag:** 1

**Summary:**

EuroBERT is a family of multilingual encoders covering English, French, German, Spanish, Chinese, Italian, Russian, Polish, Portuguese, Japanese, Vietnamese, Dutch, Arabic, Turkish, and Hindi using a dataset of 5T tokens. Model capabilities span across multilingual retrieval, sequence classification, token classification, sequence regression as well as code and math retrieval and classification tasks.

Methodological insights include:
- Higher masking ratios benefit retrieval tasks, lower ratios improve sentence classification
- Inclusion of data for code and math improves multilingual retrieval but degrades classification accuracy
- Only selecting documents with high educational value degrades performance

**Questions To Authors:**

- In Tables 9 and 10, I would suggest rephrasing "Extra-European Languages" to "Non-European Languages" to avoid confusion and reduce Eurocentricity.
- Given the large hyperparameter sweep for the larger EuroBERT, did the authors investigate applying MuP?

**Reasons To Accept:**

- The paper is comprehensively and rigorously written, with a specification of training procedure that is sufficiently detailed that one may replicate this work with relative informational ease.
- The paper has a thorough recipe analysis with several useful methodological insights.

**Reasons To Reject:**

- EuroBERT only focuses on a select few (relatively high-resource) European languages: English, German, Spanish, French, Italian, Dutch, Polish and Portugese, but misses out on the following (out of a list of the 24 official languages of the EU): Bulgarian, Croatian, Czech, Danish, Estonian, Finnish, Greek, Hungarian, Irish, Latvian, Lithuanian, Maltese, Romanian, Slovak, Slovenian, Swedish. This significantly diminishes the claims of this family of encoders being EuroBERT, and acceptance of this paper does a disservice to the multilingual and low-resource languages literature.

---

> ### Author Response · Authors · 2025-05-31
>
> Thank you for your review and suggestions. We are happy that you found that our paper provides several key insights, is clearly written and reproducible.
>
> > “EuroBERT only focuses on a select few (relatively high-resource) European languages (...) acceptance of this paper does a disservice to the multilingual and low-resource languages literature”
>
> We respectfully disagree that our paper “does a disservice to the multilingual and low-resource languages literature”: we cover a total of 15 languages, including some widely spoken but relatively low resource languages, such as Vietnamese and Hindi, and at a smaller scale Polish and Turkish. We also contribute with several insights on training multilingual encoders (for example, with extensive experimentation on the data distribution) which we believe to be valuable for the research community. While we agree that including all 24 official EU languages would be very useful, in this first iteration we chose to prioritize a smaller set of languages for which sufficient high quality data is available. Given our limited computational budget, we preferred quality over quantity, supporting only a subset of European languages and prioritizing several non-European languages in the interest of making the model more useful for the community.
>
> > “In Tables 9 and 10, I would suggest rephrasing "Extra-European Languages" to "Non-European Languages" to avoid confusion and reduce Eurocentricity.”
>
> Thank you for your suggestion. We agree that “Non-European Languages” is a clearer and more appropriate term, and we will make this change for the camera ready version.
>
> > Given the large hyperparameter sweep for the larger EuroBERT, did the authors investigate applying MuP?
>
> This is an interesting suggestion! While MuP is a promising technique for transferring the hyper-parameters from the smaller models to the bigger ones, it is primarily designed for pre-training. However, our main hyperparameter search was done for fine-tuning. Nevertheless, we consider that applying MuP at this stage is an interesting research direction for future work.
>
> We would like to thank the reviewer for their feedback and thoughtful suggestions. We will update the paper to use the term “Non-European Languages”. We also hope that our answers have addressed your concerns, but please let us know if you have any other questions that we can address during the rebuttal period, we are happy to answer them.

---

> > ### Comment · Reviewer_aExR · 2025-06-11
> >
> > Thank you for your detailed response. While I recognize the inclusion of those other low resource languages, I still maintain that the naming of EuroBERT while not including all 24 official EU languages is misleading. I will maintain my score as it is.

---

### Decision · Program_Chairs · 2025-07-08

**Decision:**

Accept

**Comment:**

The paper presents a set of multilingual encoders covering European languages and beyond. Reviewers praised the clarity of the writing and the rigor of the experimental evaluation, and overall considered the work a valuable contribution to the community. While not a major issue, I agree with reviewer aExR that the name EuroBERT may be misleading, I recommend considering an alternative name.